# On the Convergence Analysis of Muon

## Abstract

The majority of parameters in neural networks are naturally represented as matrices. However, most commonly used optimizers treat these matrix parameters as flattened vectors during optimization, potentially overlooking their inherent structural properties. Recently, an optimizer called Muon has been proposed, specifically designed to optimize matrix-structured parameters. Extensive empirical evidence shows that Muon can significantly outperform traditional optimizers when training neural networks. Nonetheless, the theoretical understanding of Muon's convergence behavior and the reasons behind its superior performance remain limited. In this work, we present a comprehensive convergence rate analysis of Muon and its comparison with Gradient Descent (GD). We characterize the conditions under which Muon can outperform GD. Our theoretical results reveal that Muon can benefit from the low-rank structure of Hessian matrices, a phenomenon widely observed in practical neural network training. Our experimental results support and corroborate the theoretical findings.

## 1 Introduction

Modern neural networks, such as large language models (LLMs) (Adler et al., 2024), typically consist of a massive amount of parameters and require significant computational resources for training. As a result, designing optimizers that can efficiently train such models has become an important and valuable research problem. Note that most of the parameters in neural networks are naturally represented as matrices, for example, weight matrices in fully connected layers, or the query, key, and value matrices in attention mechanisms. However, most widely adopted optimization algorithms such as Stochastic Gradient Descent (SGD), Adam (Kingma & Ba, 2015), and their variants often treat these matrices as flattened vectors, potentially overlooking their inherent structural properties.

Recently, Muon, an optimizer specifically designed for matrix-structured parameters, was proposed in Jordan et al. (2024). The key idea is to update the matrix parameters along an orthogonalized version of their gradients at each iteration (Algorithm 1). Empirical results across various studies and network scales consistently demonstrate the superior performance of Muon in optimizing neural networks with matrix parameters (Jordan et al., 2024; Liu et al., 2025a; An et al., 2025; Liu et al., 2025b; Ahn & Xu, 2025). However, the convergence behavior of Muon and the underlying mechanisms that contribute to its advantage over traditional gradient descent-based optimizers, such as GD and SGD, are not yet fully understood. Therefore, in this work, we aim to bridge this gap by providing a comprehensive theoretical analysis of Muon's convergence properties, along with detailed comparisons to GD-based algorithms.

In classical optimization theory, the stepsize of GD typically depends on the Lipschitz smoothness constant $L$ of the Frobenius norm, namely the maximum singular value of the Hessian. However, this quantity can be extremely large or vary rapidly during neural network training, making the stepsize of GD difficult to tune in practice. In this work, we analyze the convergence behavior of Muon not only under the standard Lipschitz smoothness constant $L$, but also under the spectral norm Lipschitz smoothness constant $L_*$, and even without assuming uniform Lipschitz smoothness. We show that the convergence of Muon can depend on $J$, which can be viewed as an average of the global Hessian information throughout training. Compared with $L$, which is determined by the maximum singular value of the Hessian, $J$ can be interpreted as averaging across both iterations and singular values. Such a smoothing effect provides potential advantages for Muon

over GD-based algorithms, especially when the Hessians exhibits low-rank structures, a phenomenon widely observed in practical neural network training. Our experimental results on quadratic regressions and neural network trainings validate these theoretical findings.

## 2 Related Work

Muon was originally proposed in Jordan et al. (2024), which demonstrated the superior performance of Muon on various models and datasets. Subsequently, Liu et al. (2025a) improved the Muon by incorporating techniques such as weight decay and adjusting the per-parameter update scale, and showed that Muon can outperform Adam when optimizing large language models (LLMs). One possible explanation for Muon's superior performance is provided by Bernstein & Newhouse (2024), which showed that Muon's update direction corresponds to the steepest descent under the spectral norm constraint. Moreover, we note that there are some recent works (Li & Hong, 2025; Pethick et al., 2025; An et al., 2025; Kovalev, 2025) that also analyze the convergence behavior of Muon. For example, Li & Hong (2025) analyzed the Frobenius norm convergence (Definition 3.4) of Muon under Frobenius norm Lipschitz smooth (Assumption 3.1). An et al. (2025) analyzed the convergence of the Simplified Muon (Algorithm 2) with Assumption 4.11. Pethick et al. (2025) introduced a general algorithmic framework based on a family of linear minimization oracle (lmo) methods and established convergence guarantees for this family. Muon can be viewed as a special instance of this framework, and its convergence was analyzed under Assumption 3.2. Kovalev (2025) proposed a stochastic non-Euclidean trust-region gradient method with momentum, treating Muon, normalized SGD, and signSGD as special cases. Kovalev (2025) established convergence guarantees for the stochastic non-Euclidean trust-region gradient method in both star-convex and nonconvex settings. Compared to Kovalev (2025), in the star-convex setting, we additionally establish the convergence of Muon with adaptive learning rates, achieving an improved complexity bound with a $\log(\epsilon^{-1})$ factor improvement over the constant-step-size result in Kovalev (2025). Recently, several works have also investigated the superiority of Muon from theoretical perspectives. Wang et al. (2025) demonstrated that Muon can outperform Adam in tail-end associative memory learning, showing that spectral orthogonalization enables more balanced updates across singular modes. Su (2025) introduced an isotropic curvature model to analyze matrix-structured optimization updates. Davis & Drusvyatskiy (2025) analyzed the squared nuclear-to-Frobenius ratio of the gradient relative to the stable rank of the incoming activations to characterize when Muon outperforms GD, and show that this condition holds in random-feature models, deep networks, and transformer training. Ma et al. (2026) analyzed matrix factorization and in-context learning of linear transformers, showing that for both problems Muon achieves iteration complexity independent of the condition number, thereby outperforming GD and Adam.

Compared to these prior works, our work provides a more comprehensive convergence analysis of Muon under various smoothness assumptions (Assumption 3.1, 3.2, 4.11), including scenarios without uniform Lipschitz smoothness (Assumption 4.13). Additionally, we offer detailed comparisons with GD and characterize the conditions under which Muon outperforms GD, and validate these conditions through experiments. Our analysis reveals that Muon can exploit the low-rank structure of Hessian matrices, offering a theoretical explanation for its advantages in structured optimization problems. Therefore, our work presents distinct contributions and insights. We believe that all these works collectively contribute to a deeper understanding of Muon.

**Low-rank and approximate blockwise diagonal structure of Hessian.** Extensive prior works (Collobert, 2004; Zhang et al., 2024b;a; Dong et al., 2025) have shown, both theoretically and empirically, that the Hessian of a neural network tends to exhibit a blockwise diagonal structure with each block corresponding to an individual neuron. Recently, Zhang et al. (2024b;a) numerically confirmed this property in small Transformers, and Dong et al. (2025) provided theoretical explanations for why such a blockwise diagonal structure emerges in neural network Hessians. In addition, many studies (Sagun et al., 2016; 2017; Wu et al., 2020; Papyan, 2020; Yao et al., 2020) have also observed that neural network Hessians are typically low-rank, i.e., their effective ranks, which can be measured by the ratio $\|H\|_*/\|H\|_{\mathrm{op}}$, can be significantly smaller than their dimensionality.

**Additional Structured Optimization Methods and Muon Variants**. Although optimizers commonly used for training deep neural networks, such as SGD and Adam, typically treat structured parameters (e.g., matrices) as flattened vectors, in recent years, there has been growing interest in designing structured optimizers that explicitly leverage the inherent structure of parameters. For instance, Adafactor (Shazeer & Stern, 2018), LAMB (You et al., 2019), and Adam-mini (Zhang et al., 2024b) incorporate matrix- or layer-level structure to reduce memory consumption. KFAC (Martens & Grosse, 2015) and TNT (Ren & Goldfarb, 2021) approximate the Fisher matrix to implement natural gradient methods. Shampoo (Gupta et al., 2018) is specifically designed for matrix or tensor parameters and can be viewed as an approximation to the full-matrix preconditioner in AdaGrad (Duchi et al., 2011). Morwani et al. (2024) provided additional theoretical analyses and modifications to Shampoo. Recently, SOAP (Vyas et al., 2024) was introduced, which combines the ideas of Adam and Shampoo. Galore (Zhao et al., 2024) was proposed to exploit the low-rank structure of gradients for memory efficiency. Additionally, Liu et al. (2025b) proposed COSMOS, which can be seen as a combination of SOAP and Muon. More recently, Xie et al. (2025) and An et al. (2025) introduced ASGO (One-Sided Shampoo), which only applies a one-sided preconditioner within the Shampoo. Xie et al. (2025); An et al. (2025) showed that ASGO can achieve better convergence rate than the original Shampoo. Crawshaw et al. (2025) provide a unified non-Euclidean steepest descent framework for neural network optimization, revealing Muon and its variants as instances of product-norm-based gradient methods and introducing more robust alternatives such as MuonMax-Momo. Liu et al. (2025c) and Qian et al. (2025) incorporate momentum variance reduction into Muon-type optimizers, achieving provably faster convergence rates and improved empirical performance.

## 3 Preliminaries

**Notation.** For a vector $v$, we denote its $l_2$ norm as $\|v\|_2$. For a matrix $A \in \mathbb{R}^{m \times n}$, we denote its nuclear norm as $\|A\|_*$, spectral norm as $\|A\|_{\text{op}}$, Frobenius norm as $\|A\|_{\text{F}}$, and $\Lambda$ norm as $\|A\|_\Lambda = \sqrt{\text{tr}(A\Lambda A^\top)}$ where $\Lambda \in \mathbb{R}^{n \times n}$ is a real symmetric positive definite matrix. For $A \in \mathbb{R}^{m \times n}$ with rows $a_1, \ldots, a_m$, we define $\overline{\text{vec}}(A) = (a_1 \, a_2 \, \cdots \, a_m)^\top \in \mathbb{R}^{mn}$ as the vectorization of $A$. We denote $I_d$ as the identity matrix with dimension $d$. For matrix $A \in \mathbb{R}^{m \times n}$, $B \in \mathbb{R}^{m' \times n'}$, we denote their Kronecker product as $A \otimes B \in \mathbb{R}^{mm' \times nn'}$. We define the operator norm of a third-order tensor $H \in \mathbb{R}^{d_1 \times d_2 \times d_3}$ as $\|H\|_{\text{op}} = \max_{\|u\|_2 = \|v\|_2 = \|w\|_2 = 1} |\sum_{i=1}^{d_1} \sum_{j=1}^{d_2} \sum_{k=1}^{d_3} H_{ijk} u_i v_j w_k|$, where $u \in \mathbb{R}^{d_1}, v \in \mathbb{R}^{d_2}, w \in \mathbb{R}^{d_3}$.

### 3.1 Optimization problem

In this paper, we consider the following stochastic optimization problem:

$$\min_{W \in \mathbb{R}^{m \times n}} f(W) = \mathbb{E}_\xi[f(W; \xi)]. \tag{1}$$

We denote $f^* = \inf_W f(W)$, $r = \min\{m, n\}$, $f_v(w) = f(W)$, and $w = \overline{\text{vec}}(W) \in \mathbb{R}^{mn}$. We assume $f^* > -\infty$. For initial point $W_0$, we denote $\Delta = f(W_0) - f^*$, and $D_F = \|W_0 - W^*\|_F$. We consider the following smoothness assumptions in this paper.

**Assumption 3.1** (Frobenius norm Lipschitz smooth). *We say $f : \mathbb{R}^{m \times n} \to \mathbb{R}$ is $L$ Frobenius norm Lipschitz smooth if for any $W, W' \in \mathbb{R}^{m \times n}$, we have*

$$\|\nabla f(W) - \nabla f(W')\|_{\text{F}} \leq L\|W - W'\|_{\text{F}}.$$

**Assumption 3.2** (Spectral norm Lipschitz smooth). *We say $f : \mathbb{R}^{m \times n} \to \mathbb{R}$ is $L_*$ spectral norm Lipschitz smooth if for any $W, W' \in \mathbb{R}^{m \times n}$, we have*

$$\|\nabla f(W) - \nabla f(W')\|_* \leq L_*\|W - W'\|_{\text{op}}.$$

Assumption 3.1 is the standard Frobenius norm-based smoothness, which serves as a natural extension of the conventional $l_2$-norm smoothness for functions with vector parameters to functions with matrix parameters. Assumption 3.2 is a spectral norm smoothness, which accounts for the distinct structure of matrix parameters. In this work, we establish the convergence of Muon under both assumptions.

For a stochastic setting, we also use the following standard bounded variance assumption.

**Assumption 3.3** (Bounded variance). *We assume $\nabla f(W; \xi)$ is an unbiased stochastic estimator of the true gradient $\nabla f(W)$ and have a bounded variance, i.e.*

$$\mathbb{E}[\nabla f(W; \xi)] = \nabla f(W), \qquad \mathbb{E}[\|\nabla f(W; \xi) - \nabla f(W)\|_{\mathrm{F}}^2] \leq \sigma^2.$$

**Definition 3.4.** We say $\widehat{W}$ is an $\epsilon$-Frobenius norm stationary point of $f$ if $\|\nabla f(\widehat{W})\|_{\mathrm{F}} \leq \epsilon$.

**Definition 3.5.** We say $\widehat{W}$ is an $\epsilon$-nuclear norm stationary point of $f$ if $\|\nabla f(\widehat{W})\|_* \leq \epsilon$.

For a matrix $A$ with rank $r$, there is a well-known relationship between its Frobenius norm and nuclear norm: $\|A\|_{\mathrm{F}} \leq \|A\|_* \leq \sqrt{r}\|A\|_{\mathrm{F}}$. Thus, we have the following proposition.

**Proposition 3.6.** *If $\widehat{W} \in \mathbb{R}^{m \times n}$ is an $\epsilon$-nuclear norm stationary point of $f$, then it is also an $\epsilon$-Frobenius norm stationary point of $f$. If $\widehat{W} \in \mathbb{R}^{m \times n}$ is an $\epsilon$-Frobenius norm stationary point of $f$, then it is also an $\sqrt{r}\epsilon$-nuclear norm stationary point of $f$, where $r = \min\{m, n\}$.*

We note that for functions with matrix parameters, the nuclear norm stationarity is tighter than the ordinary Frobenius norm stationarity.

### 3.2 Muon

---
**Algorithm 1** Muon
---
1: **Input**: Initial weights $W_0$, learning rate schedule $\{\eta_t\}$, $\beta \in [0, 1)$, batch size $B$
2: **for** $t = 0, 1, \ldots, T - 1$ **do**
3:      Sample batch $\{\xi_{t,i}\}_{i=1}^B$ uniformly
4:      $G_t = \frac{1}{B}\sum_{i=1}^B \nabla f(W_t; \xi_{t,i})$ (or $G_t = \nabla f(W_t)$ in the deterministic setting)
5:      If $t > 0$, $M_t = \beta M_{t-1} + (1 - \beta)G_t$. If $t = 0$, $M_0 = G_0$.
6:      $(U_t, S_t, V_t) = \mathrm{SVD}(M_t)$
7:      $W_{t+1} = W_t - \eta_t U_t V_t^\top$
8: **end for**

---

---
**Algorithm 2** Simplified Muon
---
1: **Input**: Initial weights $W_0$, learning rate schedule $\{\eta_t\}$
2: **for** $t = 0, 1, \ldots, T - 1$ **do**
3:      $G_t = \nabla f(W_t)$
4:      $(U_t, S_t, V_t) = \mathrm{SVD}(G_t)$
5:      $W_{t+1} = W_t - \eta_t U_t V_t^\top$
6: **end for**

---

In this subsection, we introduce the Muon algorithms. The original idea of Muon in Jordan et al. (2024) is to orthogonalize the update matrix. For example, suppose the original update direction is $G_t$ with rank $r_t$ and the singular value decomposition (SVD) of $G_t$ is $U_t S_t V_t^\top$, where $S_t \in \mathbb{R}^{r_t \times r_t}$ is the diagonal matrix of the singular values of $G_t$, $U_t \in \mathbb{R}^{m \times r_t}$ and $V_t \in \mathbb{R}^{n \times r_t}$ are the left and right singular vector matrices of $G_t$, respectively. Then, in Muon, the update matrix will be $U_t V_t^\top$, which is the nearest semi-orthogonal matrix to the original $G_t$. Following this main idea, we have Algorithm 2, which can be viewed as the simplest form of Muon. Additionally, if we add momentum and conduct orthogonalization over the momentum direction, we have Algorithm 1, which is the main algorithm we will analyze in the stochastic setting. In practice, computing the SVD is very costly. Therefore, a common practical implementation of Muon uses Newton–Schulz iterations (Bernstein & Newhouse, 2024; Jordan et al., 2024) to approximate the orthogonalization process. In this paper, we primarily analyze the theoretical properties of Algorithm 1 and 2 using SVD.

# 4 Convergence Analysis of Muon

In this section, we analyze the convergence behaviors of Muon under various smoothness assumptions and compare them with GD. The proofs of theorems in this section can be found in Appendix B and Appendix C.

## 4.1 Frobenius norm Lipschitz smooth

We first analyze the convergence of Muon with the conventional Frobenius norm Lipschitz smoothness assumption (Assumption 3.1).

**Theorem 4.1.** *Under Assumptions 3.1 and 3.3, if we apply Algorithm 1 with $\eta_t = \eta$, then*

$$\frac{1}{T} \sum_{t=0}^{T-1} \mathbb{E}[\|\nabla f(W_t)\|_*] \leq \frac{\mathbb{E}[f(W_0) - f(W_T)]}{T\eta} + \frac{Lr\eta}{2} + \frac{2\sigma\sqrt{r(1-\beta)}}{\sqrt{B(1+\beta)}} + \frac{2\sigma\sqrt{r}}{(1-\beta)T\sqrt{B}} + \frac{2r\eta\beta L}{1-\beta}.$$

By choosing the parameters in Theorem 4.1, we have following corollary.

**Corollary 4.2.** *Under the same assumptions as Theorem 4.1, we have the following convergence guarantees for Algorithm 1:*

1. **Stochastic setting:** *If we set $B = 1$, $\eta = \sqrt{\frac{(1-\beta)\Delta}{rTL}}$, and $1 - \beta = \min\left\{\frac{\sqrt{L\Delta}}{\sigma\sqrt{T}}, 1\right\}$, then*

$$\frac{1}{T} \sum_{t=0}^{T-1} \mathbb{E}[\|\nabla f(W_t)\|_*] \leq O\left(\sqrt[4]{\frac{r^2 L\Delta\sigma^2}{T}} + \sqrt{\frac{rL\Delta}{T}} + \frac{\sqrt{r}\sigma^2}{\sqrt{L\Delta T}}\right).$$

2. **Deterministic setting ($\sigma = 0$):** *If we set $\eta = \sqrt{\frac{\Delta}{rTL}}$, and $\beta = O(1)$, then*

$$\frac{1}{T} \sum_{t=0}^{T-1} \|\nabla f(W_t)\|_* \leq O\left(\sqrt{\frac{rL\Delta}{T}}\right).$$

*Therefore, the complexity to find an $\epsilon$-nuclear norm stationary point of $f$ is $O(r^2 L\sigma^2\Delta\epsilon^{-4})$ in the stochastic setting and $O(rL\Delta\epsilon^{-2})$ in the deterministic setting.*

Moreover, to study the convergence of function value of Muon, we also consider the star convex condition.

**Assumption 4.3** (Star convex). *For any $W \in \mathbb{R}^{m\times n}$, we assume the following inequality holds*

$$\langle \nabla f(W), W - W^* \rangle \geq f(W) - f^*.$$

*where $W^* \in \arg\min_W f(W)$ is the optimal point.*

Note that the star convex condition (Nesterov & Polyak, 2006) is weaker than the convex condition, and empirical evidence (Kleinberg et al., 2018; Zhou et al., 2019) suggests that, in some cases, the optimization path of neural networks can satisfy this condition.

To ensure the convergence of Muon in the star convex case, we adopt the following mild assumption. Note that the same assumption is also used in Gupta et al. (2018); An et al. (2025) for their convergence analysis of their algorithms in similar settings.

**Assumption 4.4.** *For $W_t, t = 0, \ldots, T$, generated by Algorithm 2, we assume $\|W_t - W^*\|_{\mathrm{op}} \leq D_{\mathrm{op}}$.*

In the experiment (Figure 1(b)), we can observe that when we use Muon on a quadratic function, $\|W_t - W^*\|$ is bounded, which satisfies Assumption 4.4.

With these assumptions, we have the following theorem.

**Theorem 4.5.** *Under Assumptions 4.3, 3.1, and 4.4, if we apply Algorithm 2 with $\eta_t = \eta$, then*

$$f(W_T) - f^* \leq \left(1 - \frac{\eta}{D_{\text{op}}}\right)^T (f(W_0) - f^*) + \frac{rLD_{\text{op}}\eta}{2}.$$

*Thus, to reach the precision $f(W_T) - f^* \leq \epsilon$, we can take $\eta = O(\frac{\epsilon}{rLD_{\text{op}}})$, and the complexity is $O(rLD_{\text{op}}^2\epsilon^{-1}\log\frac{\Delta}{\epsilon})$.*

**Theorem 4.6.** *Under the same assumptions as Theorem 4.5, if we apply Algorithm 2 with adaptive learning rates $\eta_t = \frac{\|\nabla f(W_t)\|_*}{rL}$, we have*

$$f(W_t) - f^* \leq \frac{2rL(f(W_0) - f^*)D_{\text{op}}^2}{2rLD_{\text{op}}^2 + t(f(W_0) - f^*)}.$$

*Thus, to reach the precision $f(W_T) - f^* \leq \epsilon$, the complexity is $O(rLD_{\text{op}}^2\epsilon^{-1})$.*

### 4.1.1 Comparison with GD

Note that in the nonconvex case, the (S)GD's complexity of finding $\epsilon$-Frobenius norm stationary points of $f$ is $O(L\sigma^2\Delta\epsilon^{-4})$ in the stochastic setting and $O(L\Delta\epsilon^{-2})$ in the deterministic setting (Garrigos & Gower, 2023), which means (S)GD can also find $\epsilon$-nuclear norm stationary points of $f$ with $O(r^2L\sigma^2\Delta\epsilon^{-4})$ complexity in the stochastic setting and $O(rL\Delta\epsilon^{-2})$ in the deterministic setting according to Proposition 3.6. Moreover, to reach the precision $f(W_T) - f^* \leq \epsilon$ in the star convex case, the complexity of GD is $O(LD_F^2\epsilon^{-1})$ (Garrigos & Gower, 2023). We have $D_F^2 = \|W_0 - W^*\|_F^2 \leq r\|W_0 - W^*\|_{\text{op}}^2 \leq rD_{\text{op}}^2$. Comparing the complexity results in Theorem 4.1 and Theorem 4.5, we can observe that under the conventional Frobenius norm Lipschitz smoothness assumption, the convergence rate of Muon is no better than that of (S)GD, and may even be worse.

## 4.2 Spectral norm Lipschitz smooth

Considering the matrix structure of the parameters and the properties of the Muon optimizer, the spectral norm Lipschitz smoothness assumption is natural in practice. In this subsection, we adopt Assumption 3.2 and establish the following theorems and corollaries.

**Theorem 4.7.** *Under Assumptions 3.2 and 3.3, if we apply Algorithm 1 with $\eta_t = \eta$, then*

$$\frac{1}{T}\sum_{t=0}^{T-1}\mathbb{E}[\|\nabla f(W_t)\|_*] \leq \frac{\mathbb{E}[f(W_0) - f(W_T)]}{T\eta} + \frac{L_*\eta}{2} + \frac{2\sigma\sqrt{r(1-\beta)}}{\sqrt{(1+\beta)B}} + \frac{2\sigma\sqrt{r}}{(1-\beta)T\sqrt{B}} + \frac{2\eta\beta L_*}{1-\beta}.$$

**Corollary 4.8.** *Under the same assumptions as Theorem 4.7, we have the following convergence guarantees for Algorithm 1:*

1. ***Stochastic setting:*** *If we set $B = 1$, $\eta = \sqrt{\frac{(1-\beta)\Delta}{TL_*}}$, and $1 - \beta = \min\left\{\frac{\sqrt{L_*\Delta}}{\sigma\sqrt{rT}}, 1\right\}$, then*

$$\frac{1}{T}\sum_{t=0}^{T-1}\mathbb{E}[\|\nabla f(W_t)\|_*] \leq O\left(\sqrt[4]{\frac{rL_*\Delta\sigma^2}{T}} + \sqrt{\frac{L_*\Delta}{T}} + \frac{r\sigma^2}{\sqrt{L_*\Delta T}}\right).$$

2. ***Deterministic setting ($\sigma = 0$):*** *If we set $\eta = \sqrt{\frac{\Delta}{TL_*}}$, and $\beta = O(1)$, then*

$$\frac{1}{T}\sum_{t=0}^{T-1}\|\nabla f(W_t)\|_* \leq O\left(\sqrt{\frac{L_*\Delta}{T}}\right).$$

*Consequently, achieving an $\epsilon$-nuclear norm stationary point of $f$ requires a complexity of $O(rL_*\sigma^2\Delta\epsilon^{-4})$ in stochastic setting and $O(L_*\Delta\epsilon^{-2})$ in deterministic setting.*

**Theorem 4.9.** *Under Assumption 4.3, 3.2 and 4.4, if we apply Algorithm 2 with $\eta_t = \eta$, then*

$$f(W_T) - f^* \le \left(1 - \frac{\eta}{D_{\mathrm{op}}}\right)^T (f(W_0) - f^*) + \frac{L_* D_{\mathrm{op}} \eta}{2}.$$

*Thus, to reach the precision $f(W_T) - f^* \le \epsilon$, we can take $\eta = O(\frac{\epsilon}{L_* D_{\mathrm{op}}})$, and the complexity is $O(L_* D_{\mathrm{op}}^2 \epsilon^{-1} log\frac{\Delta}{\epsilon})$.*

**Theorem 4.10.** *Under the same assumptions as Theorem 4.9, if we apply Algorithm 2 with adaptive learning rates $\eta_t = \frac{\|\nabla f(W_t)\|_*}{L_*}$, we have*

$$f(W_t) - f^* \le \frac{2L_*(f(W_0) - f^*)D_{\mathrm{op}}^2}{2L_* D_{\mathrm{op}}^2 + t(f(W_0) - f^*)}.$$

*Thus, to reach the precision $f(W_T) - f^* \le \epsilon$, the complexity is $O(L_* D_{\mathrm{op}}^2 \epsilon^{-1})$.*

Though we have derived the convergence theorems in terms of $L_*$, it remains unclear how to interpret them. For the Frobenius norm Lipschitz smoothness constant $L$, we know it is the largest singular value of the Hessian. The question, therefore, is what the relationship between $L$ and $L_*$ is. For a function $f$, its $L$ and $L_*$ are related to its Hessian $\nabla^2 f$. In neural network optimization, a widely observed phenomenon is that Hessians are typically low-rank (Sagun et al., 2016; 2017; Wu et al., 2020; Papyan, 2020; Yao et al., 2020) and are approximately blockwise diagonal (Dong et al., 2025; Zhang et al., 2024b;a; Collobert, 2004) (e.g., See Figure 2 in Zhang et al. (2024a) and Figure 1-3 in Dong et al. (2025)). Thus, we adopt Assumption 4.11 from An et al. (2025) to better illustrate the connections and differences between $L$ and $L_*$ in neural networks.

**Assumption 4.11** ($\Lambda$-norm Lipschitz smooth). *We say $f : \mathbb{R}^{m \times n} \to \mathbb{R}$ is 1-$\Lambda$-norm Lipschitz smooth with a positive definite matrix $\Lambda \in \mathbb{R}^{n \times n}$, if for any $W, W' \in \mathbb{R}^{m \times n}$,*

$$\|\nabla f(W) - \nabla f(W')\|_{\Lambda^{-1}} \le \|W - W'\|_{\Lambda}.$$

Assumption 4.11 is closely related to the blockwise diagonal structure of the Hessian. Actually, when $f$ is twice continuously differentiable, Assumption 4.11 is equivalent to the following assumption of Hessian:

$$-I_m \otimes \Lambda = - \begin{bmatrix} \Lambda & & & \\ & \Lambda & & \\ & & \ddots & \\ & & & \Lambda \end{bmatrix} \preceq \nabla^2 f_v(w) \preceq \begin{bmatrix} \Lambda & & & \\ & \Lambda & & \\ & & \ddots & \\ & & & \Lambda \end{bmatrix} = I_m \otimes \Lambda \in \mathbb{R}^{mn \times mn}, \qquad (2)$$

where $f_v(w) = f(W)$ and $w = \overline{\mathrm{vec}}(W) \in \mathbb{R}^{mn}$. The proof of the equivalence of these two assumptions can be found in Lemma A.3. Assumption 4.11 has also been used in An et al. (2025) to analyze matrix-structured optimization. Moreover, we have the following lemma to establish the relationship between Assumptions 3.1, 3.2, and 4.11, whose proof can be found in Appendix A.

**Lemma 4.12.** *If $f : \mathbb{R}^{m \times n} \to \mathbb{R}$ is 1-$\Lambda$-norm Lipschitz smooth with a positive definite matrix $\Lambda \in \mathbb{R}^{n \times n}$, then $f$ is also $\|\Lambda\|_{\mathrm{op}}$ Frobenius norm Lipschitz smooth and $\|\Lambda\|_*$ spectral norm Lipschitz smooth.*

### 4.2.1 Comparison with GD

In this section, we first argue that Muon inherently exploits blockwise diagonal structures and relatively low-rank properties. For example, if the Hessians of $f$ have a blockwise diagonal structure and are relatively "low rank", i.e. if it satisfies Assumption 4.11 with $\Lambda$ such that $\|\Lambda\|_{\mathrm{op}} \approx \|\Lambda\|_* \ll r\|\Lambda\|_{\mathrm{op}}$, then the convergence rate of Muon for finding the nuclear norm stationary points can be better than that of (S)GD in the nonconvex case, i.e. $O(\|\Lambda\|_* \Delta \epsilon^{-2}) \ll O(r\|\Lambda\|_{\mathrm{op}} \Delta \epsilon^{-2})$ and $O(r\|\Lambda\|_* \sigma^2 \Delta \epsilon^{-4}) \ll O(r^2 \|\Lambda\|_{\mathrm{op}} \sigma^2 \Delta \epsilon^{-4})$.

In the star convex case, recall that the complexity of GD is $O(LD_{\mathrm{F}}^2 \epsilon^{-1})$. Hence, to compare the convergence rate of Muon and GD, we need to compare $LD_{\mathrm{F}}^2$ with $L_* D_{\mathrm{op}}^2$. Similarly, if $f$ satisfies Assumption 4.11 with $\Lambda$ such that $\|\Lambda\|_{\mathrm{op}} \approx \|\Lambda\|_* \ll r\|\Lambda\|_{\mathrm{op}}$, then $L_* D_{\mathrm{op}}^2 \le \|\Lambda\|_* D_{\mathrm{op}}^2 \ll r\|\Lambda\|_{\mathrm{op}} D_{\mathrm{op}}^2$. In the experiments, we

usually find that $\|W_t - W^*\|_{\mathrm{op}}$ decreases as $t$ increases (Figure 1(b)). Thus, we assume $D_{\mathrm{op}} \approx \|W_0 - W^*\|_{\mathrm{op}}$. Despite the success of LoRA (Hu et al., 2022) in fine-tuning foundation models indicating a relatively low rank of $W_0 - W^*$ is enough for fine-tuning, many empirical evidences (Lialin et al., 2024; Jiang et al., 2024; Huang et al., 2025) demonstrate that a relatively high rank of $W_0 - W^*$ is needed for pretraining and some complex fine-tuning tasks for large foundation models, and a low rank of $W_0 - W^*$ can lead to degraded performance in those scenarios. Thus, we assume $W_0 - W^*$ is relatively "high rank". Then, we can expect $r\|W_0 - W^*\|_{\mathrm{op}}^2 \approx \|W_0 - W^*\|_{\mathrm{F}}^2$ and $L_* D_{\mathrm{op}}^2 \ll r\|\Lambda\|_{\mathrm{op}} D_{\mathrm{op}}^2 \approx L D_{\mathrm{F}}^2$, which means Muon can have a better performance than GD.

To further evaluate this argument, we conduct experiments comparing Muon and GD on a quadratic function $f(W) = \frac{1}{2}\mathrm{tr}\left((W - W^*)^\top Q(W - W^*)\right)$, where $Q$ is a positive definite matrix with a poor condition number and is relatively "low rank", i.e. $\|Q\|_* \approx \|Q\|_{\mathrm{op}}$. Note that the Hessian of $f$ is $I \otimes Q$, which satisfies Assumption 4.11. We randomly choose $W^*$, and set $W_0 = 0$. We report the results in Figure 1. We can see that Muon outperforms GD and most samples of ratio $\frac{D_{\mathrm{F}}^2 L}{D_{\mathrm{op}}^2 L_*}$ are larger than 3, which validates the theoretical findings.

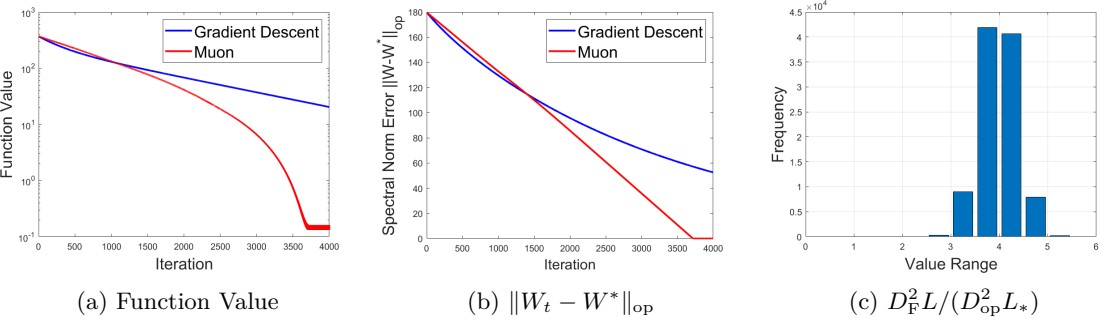

|  (a) Function Value  |  (b) $\|W_t - W^*\|_{\mathrm{op}}$  |  (c) $D_{\mathrm{F}}^2 L/(D_{\mathrm{op}}^2 L_*)$  |

Figure 1: Experiments on a quadratic function $f(W) = \frac{1}{2}\mathrm{tr}\left((W - W^*)^\top Q(W - W^*)\right)$. Detailed settings can be found in Appendix E.

Next, we consider a more realistic example to illustrate how the low-rank structure of the Hessian arises and how Muon can benefit from it. Specifically, we consider a multi-class classification problem with dataset $\{(x_i, y_i)\}_{i=1}^B$, where $x_i \in \mathbb{R}^d$ is the feature data, $y_i \in \mathbb{R}^c$ is the corresponding $c$-dimensional one-hot vector of the label, $c$ is the number of classes and $B$ is the number of samples. We first consider a linear model $g(W; x) = Wx \in \mathbb{R}^c$, where $W \in \mathbb{R}^{c \times d}$ is the parameter, and its mean-square (MSE) loss. Denote $X = (x_1, \ldots, x_B) \in \mathbb{R}^{d \times B}$, $Y = (y_1, \ldots, y_B) \in \mathbb{R}^{c \times B}$. The MSE loss can be defined as follows:

$$f(W) = \frac{1}{2B}\|WX - Y\|_{\mathrm{F}}^2. \tag{3}$$

We can calculate the Hessian of this loss as

$$\nabla^2 f_v(\overline{\mathrm{vec}}(W)) = \frac{1}{B}I_c \otimes XX^\top. \tag{4}$$

We note that this Hessian is block-diagonal with $c$ blocks and satisfy Assumption 4.11 with $\Lambda = \frac{1}{B}XX^\top$. We can calculate $L = \frac{1}{B}\|XX^\top\|_{\mathrm{op}} = \frac{1}{B}\|X\|_{\mathrm{op}}^2$ and $L_* = \frac{1}{B}\|XX^\top\|_* = \frac{1}{B}\|X\|_{\mathrm{F}}^2$.

Actually, we find that the feature matrix $X$ in practical machine learning problem is typically very "low rank", or more precisely, their singular values are highly concentrated (See Figure 2(a) and additional examples in Appendix D) such that $\|X\|_{\mathrm{op}} \approx \|X\|_{\mathrm{F}}$. For example, we randomly select 1000 samples from training datasets of MNIST (LeCun et al., 1998) and construct $X_{\mathrm{MNIST}} \in \mathbb{R}^{784 \times 1000}$. Our calculation shows that $\|X_{\mathrm{MNIST}}\|_{\mathrm{F}}^2/\|X_{\mathrm{MNIST}}\|_{\mathrm{op}}^2 = 1.41$, which means $L_* \approx L$, the Hessian of this loss is extremely low-rank. Note that this phenomenon is not limited to MNIST; similar low-rank behavior can also be observed in other real-world datasets (e.g., CIFAR-10 and text data), see Appendix D.

According to the theory, Muon can benefit from this low-rank structure and potentially converge faster than GD, thus, to verify this, we conduct the following experiments. We first use $X_{\mathrm{MNIST}}$ and their corresponding

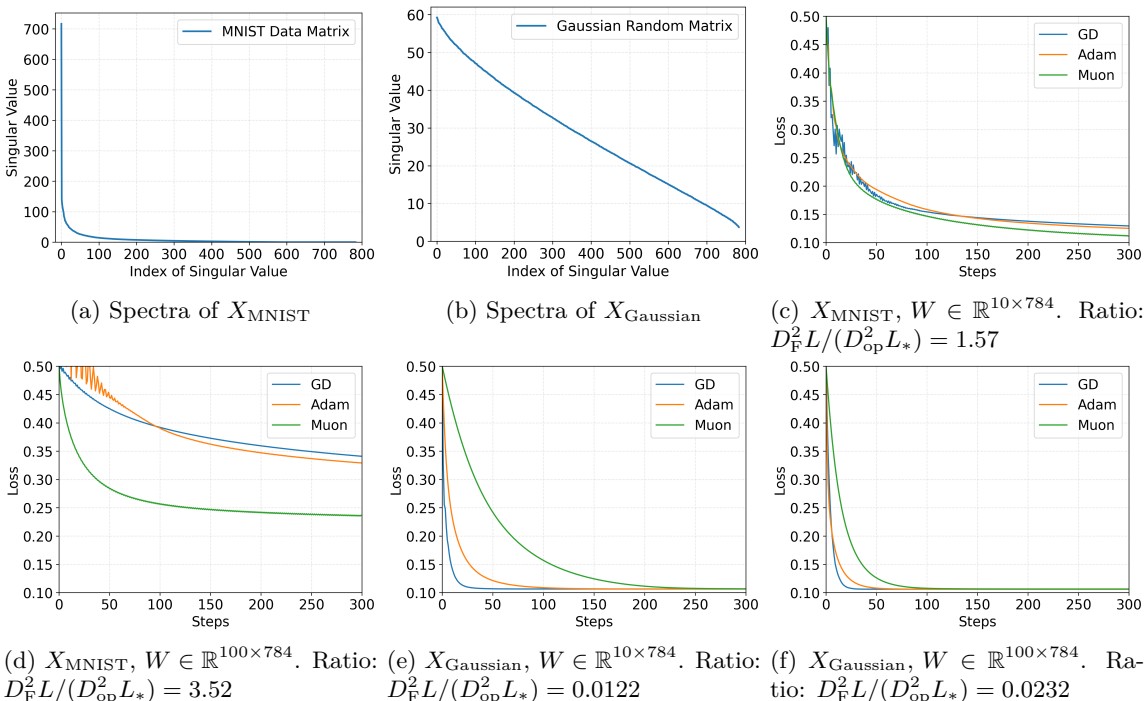

(a) Spectra of $X_{\mathrm{MNIST}}$   (b) Spectra of $X_{\mathrm{Gaussian}}$   (c) $X_{\mathrm{MNIST}}$, $W \in \mathbb{R}^{10 \times 784}$. Ratio: $D_{\mathrm{F}}^2 L / (D_{\mathrm{op}}^2 L_*) = 1.57$

(d) $X_{\mathrm{MNIST}}$, $W \in \mathbb{R}^{100 \times 784}$. Ratio: $D_{\mathrm{F}}^2 L / (D_{\mathrm{op}}^2 L_*) = 3.52$   (e) $X_{\mathrm{Gaussian}}$, $W \in \mathbb{R}^{10 \times 784}$. Ratio: $D_{\mathrm{F}}^2 L / (D_{\mathrm{op}}^2 L_*) = 0.0122$   (f) $X_{\mathrm{Gaussian}}$, $W \in \mathbb{R}^{100 \times 784}$. Ratio: $D_{\mathrm{F}}^2 L / (D_{\mathrm{op}}^2 L_*) = 0.0232$

Figure 2: (a, b): Spectra of $X_{\mathrm{MNIST}}$ and $X_{\mathrm{Gaussian}}$. (c, d, e, f): Optimizing (3) with different $X$ and $Y$.

1000 labels $y$ from the training dataset as $Y \in \mathbb{R}^{10 \times 1000}$ to optimize $W \in \mathbb{R}^{10 \times 784}$ under the MSE loss in (3). We compare GD (with/without Nesterov momentum), Adam, and Muon, tuning all hyperparameters for each method. Results in Figure 2(c) show that Muon outperforms GD and Adam. Then, we compute the Frobenius norm and operator norm of $W_0 - W^*$ (where $W^*$ is selected as the final converged point), denoted as $D_F$ and $D_{\mathrm{op}}$, respectively, and calculate the ratio $\frac{D_F^2 L}{D_{\mathrm{op}}^2 L_*}$. This ratio equals 1.57, exceeding 1 and thus supporting Muon's advantage, though it is not large, explaining the modest empirical gain in Figure 2(c). One reason is that the intrinsic dimension of $W \in \mathbb{R}^{10 \times 784}$ limits its rank to be relatively low, which in turn leads to a relatively small value of $D_F^2 / D_{\mathrm{op}}^2$. To amplify this effect, we replace $Y$ with 1000 random one-hot vectors over 100 classes, yielding $W \in \mathbb{R}^{100 \times 784}$. We then conduct the same experiment comparing GD, Adam, and Muon. The results are shown in Figure 2(d). We can observe that the ratio is now larger, reaching 3.52. Correspondingly, Muon more significantly outperforms GD and Adam in the figure, which is consistent with our theoretical predictions.

While, if we construct $X_{\mathrm{Gaussian}} \in \mathbb{R}^{784 \times 1000}$ as a Gaussian random matrix, then we can find that the singular values of this random matrix are more evenly distributed (See Figure 2(b)) and $\|X_{\mathrm{Gaussian}}\|_F^2 / \|X_{\mathrm{Gaussian}}\|_{\mathrm{op}}^2 = 224.06$, which means its $L_*$ is significantly larger than $L$ and indicating Muon will outperform worser than GD. Our experiments (Figure 2(e, f)) validated our theoretical predictions. Notably, when we increase the dimension of $y$ from 10 to 100, the corresponding ratio rises from 0.0122 to 0.0232, and the performance gap between Muon and GD (Adam) becomes smaller. As shown in Figure 2(c, d, e, f), Muon performs better relative to GD when this ratio is larger, which is consistent with our theoretical analysis. Meanwhile, our experiments also suggest that the low-rank property of the Hessian in practical machine learning problems may partially stem from the fact that real-world data are not Gaussian random, but instead exhibit some inherent low-rank structure.

### 4.3 Without uniform Lipschitz smoothness assumptions

Though the previous subsections offer some insight into how Muon can outperform GD, Assumption 4.11 appears overly restrictive. To further investigate how the structure and dynamics of the Hessian matrix affect

the convergence of Muon and GD in a more fine-grained manner, we consider the following assumption and analysis.

**Assumption 4.13.** *We assume the norm of $\nabla^3 f$ is bounded, i.e., we assume for any $W \in \mathbb{R}^{m \times n}$, $\|\nabla^3 f_v(w)\|_{\mathrm{op}} \leq s$. Here $f_v(w) = f(W)$ and $w = \overline{\mathrm{vec}}(W) \in \mathbb{R}^{mn}$.*

**Theorem 4.14.** *Under Assumption 4.13, if we apply Algorithm 2 with $\eta_t = \eta$, then*

$$\frac{1}{T} \sum_{t=0}^{T-1} \|\nabla f(W_t)\|_* \leq \frac{f(W_0) - f(W_T)}{T\eta} + \frac{\eta J}{2} + \frac{s\eta^2 r^{3/2}}{6}$$

*where $J = \frac{1}{T} \sum_{t=0}^{T-1} J_t$, $J_t = \overline{\mathrm{vec}}(U_t V_t^\top)^\top H_t \overline{\mathrm{vec}}(U_t V_t^\top)$, $H_t = \nabla^2 f_v(\overline{\mathrm{vec}}(W_t))$, and $f_v(\overline{\mathrm{vec}}(W_t)) = f(W_t)$.*

*If $J \gtrsim \sqrt{s\epsilon}\, r^{3/4}$, and $\eta = O\left(\sqrt{\frac{\Delta}{JT}}\right)$, then[1]*

$$\frac{1}{T} \sum_{t=0}^{T-1} \|\nabla f(W_t)\|_* \leq \left(\sqrt{\frac{J\Delta}{T}} + \frac{sr^{3/2}\Delta}{JT}\right).$$

Thus, it is possible that Muon can find an $\epsilon$-nuclear norm stationary point of $f$ with a complexity of $O(J\Delta\epsilon^{-2})$.

For the stochastic setting with Algorithm 1, we have the following theorem.

**Theorem 4.15.** *Under Assumptions 4.13 and 3.3, if we apply Algorithm 1 with $\eta_t = \eta$, then*

$$\frac{1}{T} \sum_{t=0}^{T-1} \mathbb{E}[\|\nabla f(W_t)\|_*] \leq \frac{\Delta}{T\eta} + \frac{J^*\eta}{2} + \frac{2\sigma\sqrt{r(1-\beta)}}{\sqrt{(1+\beta)B}} + \frac{2\sigma\sqrt{r}}{(1-\beta)T\sqrt{B}} + \frac{2\eta\beta J^*}{1-\beta} + O\left(\frac{sr^{3/2}\eta^2}{1-\beta}\right).$$

*where $J^* = \max\{\mathbb{E}[\hat{J}], \mathbb{E}[J]\}$, $\hat{J} = \frac{1}{T} \sum_{t=0}^{T-2} \hat{J}_t$, $J = \frac{1}{T} \sum_{t=0}^{T-1} J_t$, $J_t = \overline{\mathrm{vec}}(U_t V_t^\top)^\top \nabla^2 f_v(\overline{\mathrm{vec}}(W_t))\overline{\mathrm{vec}}(U_t V_t^\top)$, $\hat{J}_t = \overline{\mathrm{vec}}(U_{g,t} V_{g,t}^\top)^\top \nabla^2 f_v(\overline{\mathrm{vec}}(W_t))\overline{\mathrm{vec}}(U_t V_t^\top)$, $H_t = \nabla^2 f_v(\overline{\mathrm{vec}}(W_t))$, $f_v(\overline{\mathrm{vec}}(W_t)) = f(W_t)$, and $U_{g,t}, V_{g,t}$ are the the matrices of left and right singular vectors of $\nabla f(W_t) - \nabla f(W_{t+1})$, i.e. $\nabla f(W_t) - \nabla f(W_{t+1}) = U_{g,t} S_{g,t} V_{g,t}^\top$.*

*If $J^* \gtrsim r^{1/2}\sigma\Delta^{-1}\epsilon$, $B = 1$, $\eta = \sqrt{\frac{(1-\beta)\Delta}{TJ^*}}$, $1 - \beta = \min\{\frac{\sqrt{J^*\Delta}}{\sigma\sqrt{rT}}, 1\}$, then[1]*

$$\frac{1}{T} \sum_{t=0}^{T-1} \mathbb{E}[\|\nabla f(W_t)\|_*] \leq O\left(\sqrt[4]{\frac{rJ^*\Delta\sigma^2}{T}} + \sqrt{\frac{J^*\Delta}{T}} + \frac{r\sigma^2}{\sqrt{J^*\Delta T}} + \frac{sr^{3/2}\Delta}{TJ^*}\right).$$

Thus, it is possible that Muon can find an $\epsilon$-nuclear norm stationary point of $f$ with a complexity of $O(rJ^*\sigma^2\Delta\epsilon^{-4})$.

For the star convex case, we have

**Theorem 4.16.** *Under Assumptions 4.3, 4.13 and 4.4, if we apply Algorithm 2 with $\eta_t = \eta$, we have*

$$f(W_T) - f^* \leq \left(1 - \frac{\eta}{D_{\mathrm{op}}}\right)^T \Delta + \frac{\eta^2 \tilde{J}T}{2} + \frac{s\eta^2 D_{\mathrm{op}} r^{3/2}}{6}.$$

*where $\tilde{J} = \frac{1}{T} \sum_{t=0}^{T-1}\left(1 - \frac{\eta}{D_{\mathrm{op}}}\right)^{T-1-t} J_t$, $J_t = \overline{\mathrm{vec}}(U_t V_t^\top)^\top H_t \overline{\mathrm{vec}}(U_t V_t^\top)$, $H_t = \nabla^2 f_v(\overline{\mathrm{vec}}(W_t))$, and $f_v(\overline{\mathrm{vec}}(W_t)) = f(W_t)$.*

---

[1] In experiments, we find that usually $J_t > 0$ and is not approaching 0. Thus, we mainly discuss the situations when $J \gtrsim \sqrt{s\epsilon}\, r^{3/4}$, $J^* \gtrsim r^{1/2}\sigma\Delta^{-1}\epsilon$, and $\tilde{J} \gtrsim s^{1/2}D_{\mathrm{op}}^{-1/2}r^{3/4}\epsilon^{1/2}$ in the main paper. When $J \lesssim \sqrt{s\epsilon}\, r^{3/4}$, $J^* \lesssim r^{1/2}\sigma\Delta^{-1}\epsilon$, or $\tilde{J} \lesssim s^{1/2}D_{\mathrm{op}}^{-1/2}r^{3/4}\epsilon^{1/2}$, there is a better convergence rate regarding $\epsilon$; see discussions in Appendix B.3, B.4, and C.5.

Note that $\tilde{J}$ can be viewed as a weighted average of $J_t$ and $\tilde{J} \leq \frac{1}{T} \sum_{t=0}^{T-1} |J_t|$. If $\tilde{J} \gtrsim s^{1/2} D_{\mathrm{op}}^{-1/2} r^{3/4} \epsilon^{1/2}$ and $\eta = \min\left\{ \frac{D_{\mathrm{op}}}{T} \log\left( \frac{T\Delta}{D_{\mathrm{op}}^2 \tilde{J}} \right), D_{\mathrm{op}} \right\}$, then[1]

$$
\begin{aligned}
f(W_T) - f^* &\leq \frac{D_{\mathrm{op}}^2 \tilde{J}}{T} + \frac{D_{\mathrm{op}}^2 \tilde{J}}{2T} \left[ \log\left( \frac{T\Delta}{D_{\mathrm{op}}^2 \tilde{J}} \right) \right]^2 + \frac{s r^{3/2} D_{\mathrm{op}}^3}{6T^2} \left[ \log\left( \frac{T\Delta}{D_{\mathrm{op}}^2 \tilde{J}} \right) \right]^2 \\
&\leq \tilde{O}\left( \frac{D_{\mathrm{op}}^2 \tilde{J}}{T} \right).
\end{aligned}
$$

Thus, it is possible that Muon can reach the precision $f(W_T) - f^* \leq \epsilon$ with a complexity of $\tilde{O}(\tilde{J} D_{\mathrm{op}}^2 \epsilon^{-1})$.

### 4.3.1  Comparison with GD

Therefore, the key to comparing Muon and GD lies in comparing the relationship between $J$ (or $\tilde{J}$) and $L$. Note that $J$ is the average of $J_t$, while $\tilde{J}$ is bounded above by the average of $|J_t|$. Hence, we may first analyze $J_t$. Note that

$$
\begin{aligned}
J_t &= \overline{\mathrm{vec}}(U_t V_t^\top)^\top H_t \overline{\mathrm{vec}}(U_t V_t^\top) \\
&= \overline{\mathrm{vec}}(I_{r_t})^\top (U_t^\top \otimes V_t^\top) H_t (U_t \otimes V_t) \overline{\mathrm{vec}}(I_{r_t}) \\
&= \sum_{i,j \in [r_t]} [(U_t^\top \otimes V_t^\top) H_t (U_t \otimes V_t)]_{(i-1)*r_t+i,(j-1)*r_t+j}.
\end{aligned}
\tag{5}
$$

Thus, $J_t$ is the sum of $r_t^2$ elements of $A_t \triangleq (U_t^\top \otimes V_t^\top) H_t (U_t \otimes V_t)$, where $A_t \in \mathbb{R}^{r_t^2 \times r_t^2}$ can be viewed as a representation of $H_t$ under a certain congruence transformation depending on $U_t$ and $V_t$.

To better illustrate the relationship between $J_t$ and $L$, we assume that $H_t$ can be expressed as a Kronecker product, i.e., $H_t = P_t \otimes Q_t$, where $P_t \in \mathbb{R}^{m \times m}$ and $Q_t \in \mathbb{R}^{n \times n}$. Let $\sigma_{p,t,1} \geq \sigma_{p,t,2} \geq \cdots \geq \sigma_{p,t,m}$ denote the singular values of $P_t$, and $\sigma_{q,t,1} \geq \sigma_{q,t,2} \geq \cdots \geq \sigma_{q,t,n}$ denote the singular values of $Q_t$. Then, we have $A_t = (U_t^\top P_t U_t) \otimes (V_t^\top Q_t V_t)$ and $J_t = \langle U_t^\top P_t U_t, V_t^\top Q_t V_t \rangle \leq \sum_{i=1}^{r_t} \sigma_{p,t,i} \sigma_{q,t,i}$. Here, we use the Von Neumann's trace inequality.

Note that in the nonconvex case, the complexity of GD for $\epsilon$-nuclear norm stationary point is $O(rL\Delta\epsilon^{-2})$, where $L$ is at least larger than $\max_t L_t$ with $L_t \triangleq \|H_t\|_{\mathrm{op}} = \|P_t\|_{\mathrm{op}}\|Q_t\|_{\mathrm{op}}$. Thus, when $H_t$ can be represented by $P_t \otimes Q_t$ and $Q_t, P_t$ are relatively low-rank such that $\sum_{i=1}^{r_t} \sigma_{p,t,i}\sigma_{q,t,i} \ll r\sigma_{p,t,1}\sigma_{q,t,1}$, then $J = \frac{1}{T} \sum_t J_t \leq \frac{1}{T} \sum_t \sum_{i=1}^{r_t} \sigma_{p,t,i}\sigma_{q,t,i} \ll \frac{1}{T} \sum_t rL_t \leq rL$, and the convergence rate of Muon can be better than GD's. (Or one of $Q_t, P_t$ is relatively low-rank, i.e. if $Q_t$ is relatively low-rank such that $\|Q_t\|_{\mathrm{op}} \approx \|Q_t\| \ll r\|Q_t\|_{\mathrm{op}}$, we can also have $J = \frac{1}{T} \sum_t J_t \leq \frac{1}{T} \sum_t \|P_t\|_{\mathrm{op}}\|Q_t\|_* \ll r\max_t L_t \leq rL$.)

Similar to the previous discussions, in the star convex case, if $W_0 - W^*$ is relatively "high rank" such that $rD_{\mathrm{op}}^2 \approx \|W_0 - W^*\|_F^2$ and $Q_t, P_t$ are relatively low-rank such that $\tilde{J} \ll rL$, then we have $\tilde{J}D_{\mathrm{op}}^2 \ll LD_F^2$ and Muon can have a better performance than GD.

For stochastic case and Algorithm 1, note that $\hat{J}$ satisfies properties analogous to those of $J$. For example, if we make the same assumptions as those in Section 4.3, i.e. $H_t$ can be expressed as a Kronecker product $H_t = P_t \otimes Q_t$, where $P_t \in \mathbb{R}^{m \times m}$ and $Q_t \in \mathbb{R}^{n \times n}$. Let $\sigma_{p,t,1} \geq \sigma_{p,t,2} \geq \cdots \geq \sigma_{p,t,m}$ denote the singular values of $P_t$, and $\sigma_{q,t,1} \geq \sigma_{q,t,2} \geq \cdots \geq \sigma_{q,t,n}$ denote the singular values of $Q_t$. Then, we have $\hat{J}_t = \langle U_{g,t}^\top P_t U_t, V_{g,t}^\top Q_t V_t \rangle \leq \sum_{i=1}^{r_t} \sigma_{p,t,i}\sigma_{q,t,i}$ (Von Neumann's trace inequality). Note that in the stochastic setting, the complexity of SGD for $\epsilon$-nuclear norm stationary point is $O(r^2 L\sigma^2 \Delta\epsilon^{-4})$. Thus, Similar to the discussions in Section 4.3, when $H_t$ can be represented by $P_t \otimes Q_t$ and $Q_t, P_t$ are relatively low-rank such that $\sum_{i=1}^{r_t} \sigma_{p,t,i}\sigma_{q,t,i} \ll r\sigma_{p,t,1}\sigma_{q,t,1}$, then $J^* \leq \max\{\hat{J}, J\} \leq \max_t \sum_{i=1}^{r_t} \sigma_{p,t,i}\sigma_{q,t,i} \ll r\max_t L_t \leq rL$, and the convergence rate of Muon can be better than GD's. (Or one of $Q_t, P_t$ is relatively low-rank, i.e. if $Q_t$ is relatively low-rank such that $\|Q_t\|_{\mathrm{op}} \approx \|Q_t\| \ll r\|Q_t\|_{\mathrm{op}}$, we can also have $J^* \leq \max_t \|P_t\|_{\mathrm{op}}\|Q_t\|_* \ll r\max_t L_t \leq rL$.)

We introduce some examples to illustrate the relationship between $L$ and $J$. Similar to the last subsection, we consider the same multi-class classification dataset $\{(x_i, y_i)\}_{i=1}^B$ with a linear model $g(W; x) = Wx \in \mathbb{R}^c$,

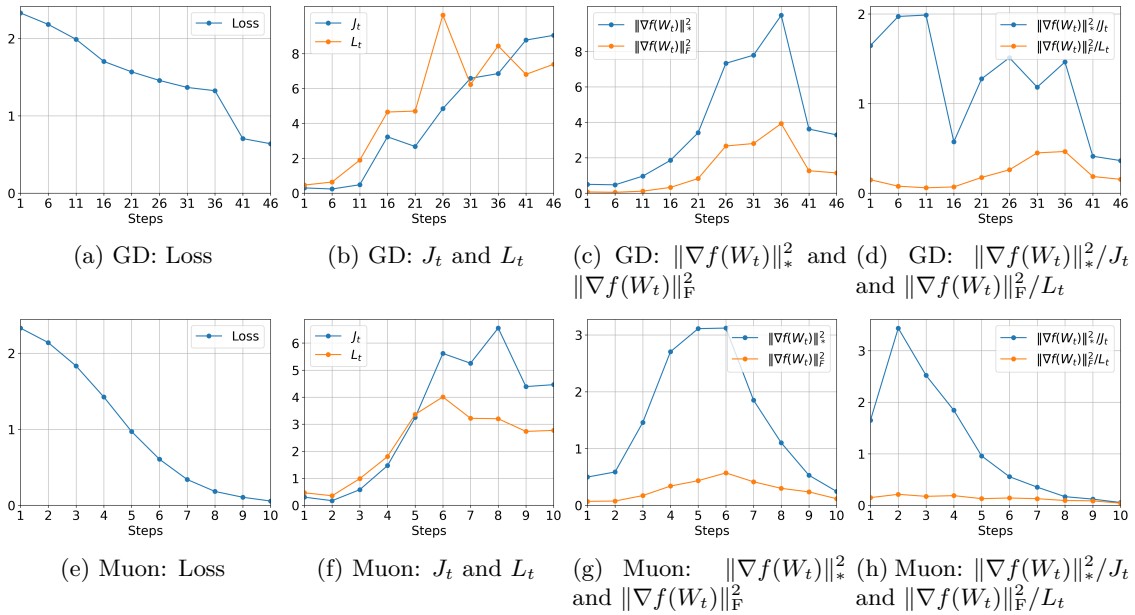

Figure 3: Comparison of $J_t$, $L_t$, $\|\nabla f(W_t)\|_*^2$ and $\|\nabla f(W_t)\|_{\mathrm{F}}^2$ over the training process of GD and Muon (Algorithm 2) on a MLP model with three matrix parameters $W^1 \in \mathbb{R}^{128 \times 784}$, $W^2 \in \mathbb{R}^{64 \times 128}$, $W^3 \in \mathbb{R}^{10 \times 64}$. We show the gradients and Hessians with respect to $W^2$ in this Figure. Detailed settings can be found in Appendix E.

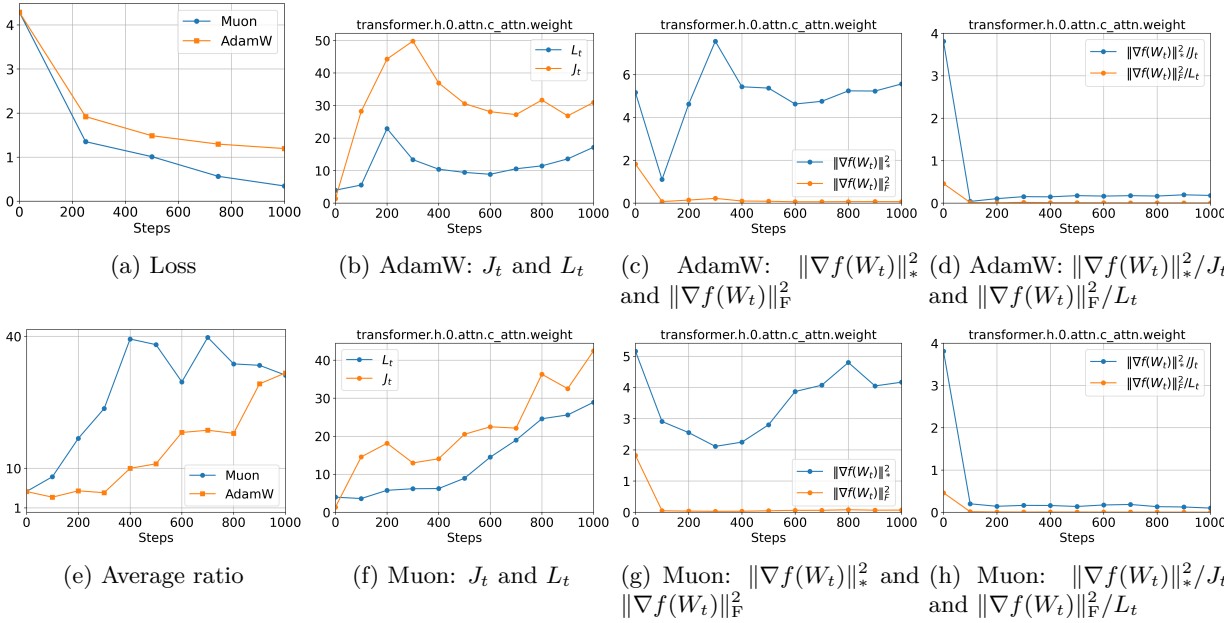

Figure 4: Comparison of $J_t$, $L_t$, $\|\nabla f(W_t)\|_*^2$, $\|\nabla f(W_t)\|_{\mathrm{F}}^2$ and the average ratio over the training process of a six-layer GPT-2 style model with AdamW and Muon. (e): The average ratio $(\|\nabla f(W_t)\|_*^2 L_t / (\|\nabla f(W_t)\|_{\mathrm{F}}^2 J_t))$ of all matrix parameters optimized by Muon. In fact, not only does the average ratio of all matrix parameters satisfy Equation (7), but every matrix parameter optimized by Muon also satisfies Equation (7) at the steps tested in our experiments. Due to space limitations, in (b, c, d, f, g, h) we present records for the parameter "transformer.h.0.attn.c_attn.weight," which is the attention matrix of the first attention layer. Detailed settings can be found in Appendix E.

where $W \in \mathbb{R}^{c \times d}$. We have discussed its MSE loss (3) in the last subsection, whose Hessian is given by $\nabla^2 f_v(\overline{\text{vec}}(W)) = \frac{1}{B} I_c \otimes X X^\top$, which has a Kronecker product structure. Thus, we have $J(\text{or } \tilde{J}, J^*) \leq \frac{1}{B} \|I_c\|_{\text{op}} \|X X^\top\|_* = \frac{1}{B} \|X\|_{\text{F}}^2 = L_*$ and the conclusion is similar to the last subsection. Here, we consider a more general case; for example, when the loss function with respect to $W$ can be represented as $f(W) = \frac{1}{B} \sum_{i=1}^{B} l(W h(x_i))$, where $h : \mathbb{R}^d \to \mathbb{R}^n$, $W \in \mathbb{R}^{m \times n}$, $l : \mathbb{R}^m \to \mathbb{R}$. Then, we have

$$\nabla^2 f_v(\overline{\text{vec}}(W)) = \frac{1}{B} \sum_{i=1}^{B} \nabla^2 l(W h(x_i)) \otimes (h(x_i) h(x_i)^\top). \tag{6}$$

Note that when $B = 1$, we have $\nabla^2 f_v(\overline{\text{vec}}(W_t)) = \nabla^2 l(W_t h(x_1)) \otimes (h(x_1) h(x_1)^\top)$, and this Hessian can be represented as a Kronecker product. Thus, we have $L \geq \max_t L_t = \|\nabla^2 l(W_t h(x_1))\|_{\text{op}} \|h(x_1) h(x_1)^\top\|_{\text{op}} = \max_t \|\nabla^2 l(W_t h(x_1))\|_{\text{op}} \|h(x_1)\|_2^2$ and $J(\text{or } \tilde{J}, J^*) \leq \frac{1}{T} \sum_{t=1}^{T} \|\nabla^2 l(W_t h(x_1))\|_{\text{op}} \|h(x_1) h(x_1)^\top\|_* = \frac{1}{T} \sum_{t=1}^{T} \|\nabla^2 l(W_t h(x_1))\|_{\text{op}} \|h(x_1)\|_2^2$. Thus, in single sample case, $J(\text{or } \tilde{J}, J^*) \leq L$, demonstrating the significant advantage of Muon compared to GD.

For multi sample case $B > 1$ and more general nonconvex loss functions, theoretical analyses of $L$ and $J$ are difficult. Here, we conduct experiments to calculate the $L_t$ and $J_t$ alongside the optimization trajectory. Moreover, various studies (Gur-Ari et al., 2018; Zhao et al., 2021; 2024; Cosson et al., 2023) have shown that the gradients during neural network optimization are also typically low rank. Thus, when we convert the stationary point of Frobenius norm to nuclear norm, the additional coefficient can be much smaller than $\sqrt{r}$. Therefore, to compare the convergence rates of Muon and GD more precisely and more fairly, one should examine the ratios between the nuclear norm and Frobenius norm of their gradients as well as $J_t$ and $L_t$. For example, we can examine the following inequality:

$$\frac{J_t}{L_t} \leq \frac{\|\nabla f(W_t)\|_*^2}{\|\nabla f(W_t)\|_{\text{F}}^2}, \tag{7}$$

If this inequality consistently holds alongside the optimization trajectory, Muon is expected to converge faster than GD.

We investigate and validate (7) during the optimization of a three-layer Multi-Layer Perceptron (MLP) trained on the MNIST dataset, as well as a six-layer GPT-2 style model trained on the Shakespeare character-level corpus. We optimize the MLP model using GD and Muon (Algorithm 2), and the GPT-2 model using AdamW and Muon. During training, we record the loss, $J_t$, $L_t$, $\|\nabla f(W_t)\|_*^2$, $\|\nabla f(W_t)\|_{\text{F}}^2$ and calculate the ratio (7). The results are shown in Figure 3 and 4, with detailed settings provided in Appendix E. In both experiments (Figure 3 and Figure 4), we observe that Muon's ratio ($\|\nabla f(W_t)\|_*^2 / J_t$) are consistently larger than GD's ($\|\nabla f(W_t)\|_{\text{F}}^2 / L_t$), demonstrating the advantage of Muon.

## 5 Conclusions and future directions

In this work, we have presented a comprehensive convergence rate analysis of Muon under various assumptions. We provided detailed comparisons between Muon and GD, and established the conditions under which Muon can outperform GD with different assumptions. We showed that Muon can exploit the low-rank structure of Hessian matrices, a property commonly observed during neural network training. Our experiments on neural networks and quadratic regressions supported and corroborated the theoretical findings. Promising directions for future research include investigating the structure of Hessian matrices from a theoretical perspective and leveraging these structural properties to develop new or more effective optimization methods.

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

## Appendix

The Appendix is organized as follows. In Appendix A, we introduce some lemmas that will be utilized in the subsequent proofs, and we also give the proofs of Lemma 4.12 and Lemma A.3. In Appendix B, we present the proofs of theorems in the nonconvex setting. In Appendix C, we present the proofs of theorems in the star convex setting. In Appendix D, we present additional examples that illustrate the low-rank structure of real-world data matrices. In Appendix E, we provide the detailed settings of our experiments.

## A   Lemmas

**Lemma A.1** (Lemma 8 in An et al. (2025))**.** *For a symmetric positive definite matrix* $\Lambda \in \mathbb{R}^{n \times n}$ *and matrix* $A \in \mathbb{R}^{m \times n}$*, it holds that*

$$\|A\|_* \leq \sqrt{\|\Lambda\|_*} \|A\|_{\Lambda^{-1}}.$$

**Lemma A.2.** *For a symmetric positive definite matrix* $\Lambda \in \mathbb{R}^{n \times n}$ *and matrix* $A \in \mathbb{R}^{m \times n}$*, it holds that*

$$\|A\|_{\mathrm{F}} \leq \sqrt{\|\Lambda\|_{\mathrm{op}}} \|A\|_{\Lambda^{-1}}, \tag{8}$$

$$\|A\|_\Lambda \leq \sqrt{\|\Lambda\|_{\mathrm{op}}} \|A\|_{\mathrm{F}}, \tag{9}$$

$$\|A\|_\Lambda \leq \sqrt{\|\Lambda\|_*} \|A\|_{\mathrm{op}}. \tag{10}$$

*Proof.* We first prove Equation (8) and Equation (9).

Suppose the spectral decomposition of $\Lambda$ is $\Lambda = U\Sigma U^\top$, where $U$ is an orthogonal matrix, $\Sigma = \mathrm{diag}(\sigma_1, \sigma_2, \ldots, \sigma_n)$ with $\sigma_1 \geq \sigma_2 \geq \cdots \geq \sigma_n$ being the ordered singular values of $\Lambda$. Denote $AU = B = (b_1\, b_2\, \ldots\, b_n)$, where $b_i$ is the $i$-th column of $B$. Then, we have

$$\|A\|_\Lambda^2 = \mathrm{tr}\left(A\Lambda A^\top\right) = \mathrm{tr}\left(AU\Sigma U^\top A^\top\right) = \mathrm{tr}\left(\Sigma B^\top B\right) = \sum_{i=1}^n \sigma_i \|b_i\|_2^2.$$

Note that

$$\|A\|_{\mathrm{F}}^2 = \sum_{i=1}^n \|b_i\|_2^2.$$

Thus, we have

$$\|A\|_\Lambda \leq \sqrt{\sigma_1} \|A\|_{\mathrm{F}} = \sqrt{\|\Lambda\|_{\mathrm{op}}} \|A\|_{\mathrm{F}},$$

and

$$\|A\|_\Lambda \geq \sqrt{\sigma_n} \|A\|_{\mathrm{F}}.$$

Note that $\sigma_n^{-1} = \left\|\Lambda^{-1}\right\|_{\mathrm{op}}$. Thus,

$$\|A\|_{\mathrm{F}} \leq \sqrt{\sigma_n^{-1}} \|A\|_\Lambda = \sqrt{\|\Lambda^{-1}\|_{\mathrm{op}}} \|A\|_\Lambda. \tag{11}$$

Since Equation (11) holds for any $\Lambda$, we can define $\Lambda' = \Lambda^{-1}$ and get

$$\|A\|_{\mathrm{F}} \leq \sqrt{\|\Lambda'\|_{\mathrm{op}}} \|A\|_{\Lambda'^{-1}}. \tag{12}$$

Thus, we have proved Equation (8) and Equation (9). For Equation (10), we can prove it with the following inequalities

$$\|A\|_\Lambda = \sqrt{\mathrm{tr}\left(A\Lambda A^\top\right)} = \sqrt{\mathrm{tr}\left(A^\top A\Lambda\right)} \leq \sqrt{\|A^\top A\|_{\mathrm{op}} \|\Lambda\|_*} = \sqrt{\|\Lambda\|_*} \|A\|_{\mathrm{op}}.$$

$$\square$$

**Lemma A.3.** *Let $\Lambda \in \mathbb{R}^{n \times n}$ be positive definite. For a twice continuously differentiable function $f : \mathbb{R}^{m \times n} \to \mathbb{R}$, define $f_v(w) = f(W)$ with $w = \overline{\mathrm{vec}}(W) \in \mathbb{R}^{mn}$. Then the following are equivalent:*

    *1. $f$ satisfies Assumption 4.11, i.e. for all $W, W' \in \mathbb{R}^{m \times n}$,*

$$\|\nabla f(W) - \nabla f(W')\|_{\Lambda^{-1}} \leq \|W - W'\|_{\Lambda}.$$

    *2. For all $w \in \mathbb{R}^{mn}$,*

$$-I_m \otimes \Lambda \preceq \nabla^2 f_v(w) \preceq I_m \otimes \Lambda.$$

*Proof.* Let $M := I_m \otimes \Lambda$. For any $A \in \mathbb{R}^{m \times n}$, we have

$$\mathrm{tr}\left(A\Lambda A^\top\right) = \overline{\mathrm{vec}}(A)^\top (I_m \otimes \Lambda)\overline{\mathrm{vec}}(A) = \overline{\mathrm{vec}}(A)^\top M \overline{\mathrm{vec}}(A),$$

and similarly

$$\mathrm{tr}\left(G\Lambda^{-1}G^\top\right) = \overline{\mathrm{vec}}(G)^\top M^{-1}\overline{\mathrm{vec}}(G).$$

Moreover,

$$\nabla f_v(w) = \overline{\mathrm{vec}}(\nabla f(W)).$$

Hence the first condition is equivalent to

$$\|\nabla f_v(w) - \nabla f_v(w')\|_{M^{-1}} \leq \|w - w'\|_M, \qquad \forall w, w' \in \mathbb{R}^{mn}.$$

Here, for a vector $v \in \mathbb{R}^d$, its norm $\|v\|_A$ is defined as $\|v\|_A := \sqrt{v^\top A v} = \|A^{1/2}v\|_2$, where $A \in \mathbb{R}^{d \times d}$ is positive definite.

We first prove $(2) \Rightarrow (1)$. For any $w, w' \in \mathbb{R}^{mn}$, we have

$$\nabla f_v(w) - \nabla f_v(w') = \int_0^1 \nabla^2 f_v(w' + t(w - w'))) (w - w')\, dt.$$

Therefore,

$$M^{-1/2}(\nabla f_v(w) - \nabla f_v(w')) = \int_0^1 \left(M^{-1/2}\nabla^2 f_v(w' + t(w - w'))M^{-1/2}\right)M^{1/2}(w - w')\, dt.$$

Under (2), we have

$$-I \preceq M^{-1/2}\nabla^2 f_v(\cdot)M^{-1/2} \preceq I,$$

so its operator norm is at most 1. Thus

$$
\begin{aligned}
\|M^{-1/2}(\nabla f_v(w) - \nabla f_v(w'))\|_2 &= \left\|\int_0^1 \left(M^{-1/2}\nabla^2 f_v(w' + t(w - w'))M^{-1/2}\right)M^{1/2}(w - w')\, dt\right\|_2 \\
&\leq \int_0^1 \left\|M^{-1/2}\nabla^2 f_v(w' + t(w - w'))M^{-1/2}\right\|_{\mathrm{op}} \|M^{1/2}(w - w')\|_2\, dt \\
&\leq \|M^{1/2}(w - w')\|_2,
\end{aligned}
$$

which means

$$\|\nabla f_v(w) - \nabla f_v(w')\|_{M^{-1}} \leq \|w - w'\|_M.$$

Next we prove $(1) \Rightarrow (2)$. Fix any $w$ and any direction $u \in \mathbb{R}^{mn}$. Applying (1) to $w + tu$ and $w$ gives

$$\|\nabla f_v(w + tu) - \nabla f_v(w)\|_{M^{-1}} \leq |t| \, \|u\|_M.$$

Dividing by $|t|$ and letting $t \to 0$ yields

$$\|\nabla^2 f_v(w)u\|_{M^{-1}} \leq \|u\|_M, \qquad \forall u.$$

Let $v := M^{1/2}u$, we have $u = M^{-1/2}v$ and

$$\|M^{-1/2}\nabla^2 f_v(w)M^{-1/2}v\|_2 \leq \|v\|_2,$$

which implies

$$\|M^{-1/2}\nabla^2 f_v(w)M^{-1/2}\|_{\mathrm{op}} \leq 1.$$

Since $\nabla^2 f_v(w)$ is symmetric, this is equivalent to

$$-I \preceq M^{-1/2}\nabla^2 f_v(w)M^{-1/2} \preceq I.$$

Multiplying both sides by $M^{1/2}$ proves

$$-M \preceq \nabla^2 f_v(w) \preceq M,$$

i.e.,

$$-I_m \otimes \Lambda \preceq \nabla^2 f_v(w) \preceq I_m \otimes \Lambda.$$

This completes the proof. □

**Proof of Lemma 4.12**

*Proof.* If $f : \mathbb{R}^{m \times n} \to \mathbb{R}$ is 1-$\Lambda$-norm Lipschitz smooth with a positive definite matrix $\Lambda \in \mathbb{R}^{n \times n}$, then, for any $W, W' \in \mathbb{R}^{m \times n}$, we have

$$\begin{aligned}
\|\nabla f(W) - \nabla f(W')\|_{\mathrm{F}} &\leq \sqrt{\|\Lambda\|_{\mathrm{op}}}\|\nabla f(W) - \nabla f(W')\|_{\Lambda^{-1}} \\
&\leq \sqrt{\|\Lambda\|_{\mathrm{op}}}\|W - W'\|_{\Lambda} \\
&\leq \|\Lambda\|_{\mathrm{op}}\|W - W'\|_{\mathrm{F}}
\end{aligned}$$

where the first inequality is due to Equation (8) in Lemma A.2, the second inequality is the definition of the $\Lambda$-smoothness, and the third inequality is due to Equation (9) in Lemma A.2. Thus, $f$ is also $\|\Lambda\|_{\mathrm{op}}$ Frobenius norm Lipschitz smooth.

Moreover, we have

$$\begin{aligned}
\|\nabla f(W) - \nabla f(W')\|_* &\leq \sqrt{\|\Lambda\|_*}\|\nabla f(W) - \nabla f(W')\|_{\Lambda^{-1}} \\
&\leq \sqrt{\|\Lambda\|_*}\|W - W'\|_{\Lambda} \\
&\leq \|\Lambda\|_*\|W - W'\|_{\mathrm{op}}
\end{aligned}$$

where the first inequality is due to Lemma A.1, the second inequality is the definition of the $\Lambda$-smoothness, and the third inequality is due to Equation (10) in Lemma A.2. Thus, $f$ is also $\|\Lambda\|_*$ spectral norm Lipschitz smooth. □

**Lemma A.4.** *For $t = 0, 1, \ldots, T$, $M_t$ and $W_t$ generated by Algorithm 1, defining $C_0 = \nabla f(W_0)$, and when $t > 0$, $C_t = \beta C_{t-1} + (1 - \beta)\nabla f(W_t) = (1 - \beta)\sum_{i=1}^{t}\beta^{t-i}\nabla f(W_i) + \beta^t \nabla f(W_0)$, we have*

$$\mathbb{E}[\|C_t - M_t\|_{\mathrm{F}}] \leq \sqrt{\frac{1 - \beta}{1 + \beta}}\frac{\sigma}{\sqrt{B}} + \beta^t \frac{\sigma}{\sqrt{B}}.$$

*Proof.* According to Assumption 3.3, we have the following relationship about $G_t$ and $\nabla f(W_t)$.

$$\begin{aligned}
\mathbb{E}[\|G_t - \nabla f(W_t)\|_{\mathrm{F}}^2] &= \mathbb{E}\left[\left\|\frac{1}{B}\sum_{i=1}^{B}\nabla f(W_t; \xi_{t,i}) - \nabla f(W_t)\right\|_{\mathrm{F}}^2\right] \\
&= \frac{1}{B^2}\sum_{i=1}^{B}\mathbb{E}[\|\nabla f(W_t; \xi_{t,i}) - \nabla f(W_t)\|_{\mathrm{F}}^2]
\end{aligned}$$

$$\leq \frac{\sigma^2}{B} \tag{13}$$

where the second equality is due to $\mathbb{E}[\nabla f(W_t; \xi_{t,i})] = \nabla f(W_t)$ and the last inequality is due to $\mathbb{E}\|\nabla f(W; \xi_{t,i}) - \nabla f(W)\|_{\mathrm{F}}^2 \leq \sigma^2$. Moreover, according to the Cauchy-Schwarz inequality, we have

$$\mathbb{E}[\|G_t - \nabla f(W_t)\|_{\mathrm{F}}] \leq \sqrt{\mathbb{E}[\|G_t - \nabla f(W_t)\|_{\mathrm{F}}^2]} = \frac{\sigma}{\sqrt{B}}. \tag{14}$$

Note that $M_0 = G_0$, $M_t = \beta M_{t-1} + (1-\beta)G_t = (1-\beta)\sum_{i=1}^t \beta^{t-i}G_i + \beta^t G_0$. Thus, we have

$$
\begin{aligned}
\mathbb{E}[\|C_t - M_t\|_{\mathrm{F}}] \leq & (1-\beta)\mathbb{E}[\|\sum_{i=1}^t \beta^{t-i}(G_i - \nabla f(W_i))\|_{\mathrm{F}}] + \beta^t \mathbb{E}[\|G_0 - \nabla f(W_0)\|_{\mathrm{F}}] \\
\leq & \sqrt{(1-\beta)^2 \mathbb{E}[\|\sum_{i=1}^t \beta^{t-i}(G_i - \nabla f(W_i))\|_{\mathrm{F}}^2] + \beta^t \frac{\sigma}{\sqrt{B}}} \\
\leq & \sqrt{(1-\beta)^2 \sum_{i=1}^t \beta^{2t-2i}\frac{\sigma^2}{B} + \beta^t \frac{\sigma}{\sqrt{B}}} \\
\leq & \sqrt{\frac{1-\beta}{1+\beta}}\frac{\sigma}{\sqrt{B}} + \beta^t \frac{\sigma}{\sqrt{B}}
\end{aligned}
$$

where the second inequality is due to the Cauchy-Schwarz inequality and Equation (14), the third inequality is due to Equation (13). $\qquad\square$

## B  Nonconvex

### B.1  Proof of Theorem 4.1

*Proof.* Set $\eta_t = \eta$. Since $f$ is $L$ Frobenius norm Lipschitz smooth, we have

$$
\begin{aligned}
\mathbb{E}[f(W_t) - f(W_{t+1})] \geq & \mathbb{E}[\eta\langle \nabla f(W_t), U_t V_t^\top \rangle - \frac{L}{2}\eta^2 \|U_t V_t^\top\|_{\mathrm{F}}^2] \\
\geq & \mathbb{E}[\eta\langle M_t, U_t V_t^\top \rangle - \frac{rL}{2}\eta^2 - \eta\langle \nabla f(W_t) - M_t, U_t V_t^\top \rangle] \\
\geq & \mathbb{E}[\eta\langle M_t, U_t V_t^\top \rangle - \frac{rL}{2}\eta^2 - \eta\|\nabla f(W_t) - M_t\|_{\mathrm{F}}\|U_t V_t^\top\|_{\mathrm{F}}] \\
\geq & \mathbb{E}[\eta\|M_t\|_* - \frac{rL}{2}\eta^2 - \eta\sqrt{r}\|\nabla f(W_t) - M_t\|_{\mathrm{F}}] \\
\geq & \mathbb{E}[\eta\|\nabla f(W_t)\|_* - \frac{rL}{2}\eta^2 - 2\eta\sqrt{r}\|\nabla f(W_t) - M_t\|_{\mathrm{F}}].
\end{aligned}
$$

Then, we need to analyze and bound the error $\|\nabla f(W_t) - M_t\|_{\mathrm{F}}$. Note that $M_0 = G_0$, $M_t = \beta M_{t-1} + (1-\beta)G_t = (1-\beta)\sum_{i=1}^t \beta^{t-i}G_i + \beta^t G_0$. We can define $C_0 = \nabla f(W_0)$, and when $t > 0$, $C_t = \beta C_{t-1} + (1-\beta)\nabla f(W_t) = (1-\beta)\sum_{i=1}^t \beta^{t-i}\nabla f(W_i) + \beta^t \nabla f(W_0)$. Then, according to Lemma A.4, we have

$$\mathbb{E}[\|C_t - M_t\|_{\mathrm{F}}] \leq \sqrt{\frac{1-\beta}{1+\beta}}\frac{\sigma}{\sqrt{B}} + \beta^t \frac{\sigma}{\sqrt{B}}.$$

Moreover, when $t > 0$, we note that

$$
\begin{aligned}
& \mathbb{E}[\|\nabla f(W_t) - C_t\|_{\mathrm{F}}] \\
= & \mathbb{E}[\|\nabla f(W_t) - (\beta C_{t-1} + (1-\beta)\nabla f(W_t))\|_{\mathrm{F}}]
\end{aligned}
$$

$$
\begin{aligned}
&= \mathbb{E}[\beta \|\nabla f(W_t) - C_{t-1}\|_{\mathrm{F}}] \\
&\leq \mathbb{E}[\beta \|\nabla f(W_{t-1}) - C_{t-1}\|_{\mathrm{F}} + \beta \|\nabla f(W_{t-1}) - \nabla f(W_t)\|_{\mathrm{F}}] \\
&\leq \mathbb{E}[\beta \|\nabla f(W_{t-1}) - C_{t-1}\|_{\mathrm{F}} + \beta L \|W_{t-1} - W_t\|_{\mathrm{F}}] \\
&= \mathbb{E}[\beta \|\nabla f(W_{t-1}) - C_{t-1}\|_{\mathrm{F}} + \beta L \eta \|U_{t-1} V_{t-1}^\top\|_{\mathrm{F}}] \\
&\leq \mathbb{E}[\beta \|\nabla f(W_{t-1}) - C_{t-1}\|_{\mathrm{F}} + \beta L \eta \sqrt{r}] \\
&\leq \beta^t \|\nabla f(W_0) - C_0\|_{\mathrm{F}} + \sum_{i=1}^{t} \beta^i L \eta \sqrt{r} \\
&\leq \frac{\sqrt{r}\beta L \eta}{1 - \beta}
\end{aligned}
$$

where the second inequality is due to Assumption 3.1.

Thus, we have

$$
\mathbb{E}[\|\nabla f(W_t) - M_t\|_{\mathrm{F}}] \leq \mathbb{E}[\|C_t - M_t\|_{\mathrm{F}} + \|\nabla f(W_t) - C_t\|_{\mathrm{F}}] \leq \sqrt{\frac{1-\beta}{1+\beta}} \frac{\sigma}{\sqrt{B}} + \beta^t \frac{\sigma}{\sqrt{B}} + \frac{\sqrt{r}\beta L \eta}{1-\beta},
$$

and

$$
\begin{aligned}
\mathbb{E}[f(W_t) - f(W_{t+1})] &\geq \mathbb{E}[\eta \|\nabla f(W_t)\|_* - \frac{rL}{2}\eta^2 - 2\eta\sqrt{r}\|\nabla f(W_t) - M_t\|_{\mathrm{F}}] \\
&\geq \mathbb{E}[\eta \|\nabla f(W_t)\|_* - \frac{rL}{2}\eta^2 - 2\eta\sqrt{\frac{1-\beta}{1+\beta}}\frac{\sigma\sqrt{r}}{\sqrt{B}} - 2\eta\beta^t\frac{\sigma\sqrt{r}}{\sqrt{B}} - \frac{2r\eta^2\beta L}{1-\beta}].
\end{aligned}
$$

Summing over $t = 0, 1, \ldots, T-1$, we can get

$$
\frac{1}{T}\sum_{t=0}^{T-1} \mathbb{E}[\|\nabla f(W_t)\|_*] \leq \frac{1}{T\eta}\mathbb{E}[f(W_0) - f(W_T)] + \frac{Lr\eta}{2} + \frac{2\sigma\sqrt{r(1-\beta)}}{\sqrt{(1+\beta)B}} + \frac{2\sigma\sqrt{r}}{(1-\beta)T\sqrt{B}} + \frac{2r\eta\beta L}{1-\beta}.
$$

Set $\eta = \sqrt{\frac{(1-\beta)\Delta}{rTL}}$, we can get

$$
\frac{1}{T}\sum_{t=0}^{T-1} \mathbb{E}[\|\nabla f(W_t)\|_*] \leq \frac{\Delta}{T\eta} + \frac{Lr\eta}{2} + \frac{2\sigma\sqrt{r(1-\beta)}}{\sqrt{(1+\beta)B}} + \frac{2\sigma\sqrt{r}}{(1-\beta)T\sqrt{B}} + \frac{2r\eta\beta L}{1-\beta}.
$$

When $B = 1$, we can set $1 - \beta = \min\{\frac{\sqrt{L\Delta}}{\sigma\sqrt{T}}, 1\}$ and get

$$
\frac{1}{T}\sum_{t=0}^{T-1} \mathbb{E}[\|\nabla f(W_t)\|_*] \leq O(1)\left( \sqrt[4]{\frac{r^2 L \Delta \sigma^2}{T}} + \sqrt{\frac{rL\Delta}{T}} + \frac{\sqrt{r}\sigma^2}{\sqrt{L\Delta T}} \right).
$$

Thus, we can find an $\epsilon$-nuclear norm stationary point of $f$ with a complexity of $O(r^2 L \sigma^2 \Delta \epsilon^{-4})$. $\qquad\square$

## B.2   Proof of Theorem 4.7

*Proof.* Set $\eta_t = \eta$. Since $f$ is $L_*$ spectral norm Lipschitz smooth, we have

$$
\begin{aligned}
\mathbb{E}[f(W_t) - f(W_{t+1})] &\geq \mathbb{E}[\eta\langle \nabla f(W_t), U_t V_t^\top \rangle - \frac{L_*}{2}\eta^2 \|U_t V_t^\top\|_{\mathrm{op}}^2] \\
&\geq \mathbb{E}[\eta\langle M_t, U_t V_t^\top \rangle - \frac{L_*}{2}\eta^2 - \eta\langle \nabla f(W_t) - M_t, U_t V_t^\top \rangle] \\
&\geq \mathbb{E}[\eta\langle M_t, U_t V_t^\top \rangle - \frac{L_*}{2}\eta^2 - \eta\|\nabla f(W_t) - M_t\|_* \|U_t V_t^\top\|_{\mathrm{op}}]
\end{aligned}
$$

$$\geq \mathbb{E}[\eta\|M_t\|_* - \frac{L_*}{2}\eta^2 - \eta\|\nabla f(W_t) - M_t\|_*]$$

$$\geq \mathbb{E}[\eta\|\nabla f(W_t)\|_* - \frac{L_*}{2}\eta^2 - 2\eta\|\nabla f(W_t) - M_t\|_*].$$

Then, we need to analyze and bound the error $\|\nabla f(W_t) - M_t\|_*$. Note that $M_0 = G_0$, $M_t = \beta M_{t-1} + (1-\beta)G_t = (1-\beta)\sum_{i=1}^{t}\beta^{t-i}G_i + \beta^t G_0$. We can define $C_0 = \nabla f(W_0)$, and when $t > 0$, $C_t = \beta C_{t-1} + (1-\beta)\nabla f(W_t) = (1-\beta)\sum_{i=1}^{t}\beta^{t-i}\nabla f(W_i) + \beta^t \nabla f(W_0)$. Then, according to Lemma A.4, we have

$$\mathbb{E}[\|C_t - M_t\|_*] \leq \sqrt{r}\mathbb{E}[\|C_t - M_t\|_{\mathrm{F}}] \leq \sqrt{\frac{1-\beta}{1+\beta}}\frac{\sigma\sqrt{r}}{\sqrt{B}} + \beta^t \frac{\sigma\sqrt{r}}{\sqrt{B}}.$$

Moreover, when $t > 0$, we can note that

$$\mathbb{E}[\|\nabla f(W_t) - C_t\|_*]$$
$$= \mathbb{E}[\|\nabla f(W_t) - (\beta C_{t-1} + (1-\beta)\nabla f(W_t))\|_*]$$
$$= \mathbb{E}[\beta\|\nabla f(W_t) - C_{t-1}\|_*]$$
$$\leq \mathbb{E}[\beta\|\nabla f(W_{t-1}) - C_{t-1}\|_* + \beta\|\nabla f(W_{t-1}) - \nabla f(W_t)\|_*]$$
$$\leq \mathbb{E}[\beta\|\nabla f(W_{t-1}) - C_{t-1}\|_* + \beta L_*\|W_{t-1} - W_t\|_{\mathrm{op}}]$$
$$= \mathbb{E}[\beta\|\nabla f(W_{t-1}) - C_{t-1}\|_* + \beta L_*\eta\|U_{t-1}V_{t-1}^{\top}\|_{\mathrm{op}}]$$
$$\leq \mathbb{E}[\beta\|\nabla f(W_{t-1}) - C_{t-1}\|_* + \beta L_*\eta]$$
$$\leq \beta^t \|\nabla f(W_0) - C_0\|_{\mathrm{F}} + \sum_{i=1}^{t}\beta^i L_*\eta$$
$$\leq \frac{\beta L_*\eta}{1-\beta}$$

where the second inequality is due to Assumption 3.2.

Thus, we have

$$\mathbb{E}[\|\nabla f(W_t) - M_t\|_*] \leq \mathbb{E}[\|C_t - M_t\|_* + \|\nabla f(W_t) - C_t\|_*] \leq \sqrt{\frac{1-\beta}{1+\beta}}\frac{\sigma\sqrt{r}}{\sqrt{B}} + \beta^t \frac{\sigma\sqrt{r}}{\sqrt{B}} + \frac{\beta L_*\eta}{1-\beta},$$

and

$$\mathbb{E}[f(W_t) - f(W_{t+1})] \geq \mathbb{E}[\eta\|\nabla f(W_t)\|_* - \frac{L_*}{2}\eta^2 - 2\eta\|\nabla f(W_t) - M_t\|_*]$$

$$\geq \mathbb{E}[\eta\|\nabla f(W_t)\|_* - \frac{L_*}{2}\eta^2 - 2\eta\sqrt{\frac{1-\beta}{1+\beta}}\frac{\sigma\sqrt{r}}{\sqrt{B}} - 2\eta\beta^t \frac{\sigma\sqrt{r}}{\sqrt{B}} - \frac{2\eta^2\beta L_*}{1-\beta}].$$

Summing over $t = 0, 1, \ldots, T-1$, we can get

$$\frac{1}{T}\sum_{t=0}^{T-1}\mathbb{E}[\|\nabla f(W_t)\|_*] \leq \frac{1}{T\eta}\mathbb{E}[f(W_0) - f(W_T)] + \frac{L_*\eta}{2} + \frac{2\sigma\sqrt{r(1-\beta)}}{\sqrt{(1+\beta)B}} + \frac{2\sigma\sqrt{r}}{(1-\beta)T\sqrt{B}} + \frac{2\eta\beta L_*}{1-\beta}.$$

Set $\eta = \sqrt{\frac{(1-\beta)\Delta}{TL_*}}$, we can get

$$\frac{1}{T}\sum_{t=0}^{T-1}\mathbb{E}[\|\nabla f(W_t)\|_*] \leq \frac{\Delta}{T\eta} + \frac{L_*\eta}{2} + \frac{2\sigma\sqrt{r(1-\beta)}}{\sqrt{(1+\beta)B}} + \frac{2\sigma\sqrt{r}}{(1-\beta)T\sqrt{B}} + \frac{2\eta\beta L_*}{1-\beta}.$$

When $B = 1$, we can set $1 - \beta = \min\{\frac{\sqrt{L_*\Delta}}{\sigma\sqrt{rT}}, 1\}$ and get

$$\frac{1}{T}\sum_{t=0}^{T-1}\mathbb{E}[\|\nabla f(W_t)\|_*] \leq O(1)\left(\sqrt[4]{\frac{rL_*\Delta\sigma^2}{T}} + \sqrt{\frac{L_*\Delta}{T}} + \frac{r\sigma^2}{\sqrt{L_*\Delta T}}\right).$$

Thus, we can find an $\epsilon$-nuclear norm stationary point of $f$ with a complexity of $O(rL_*\sigma^2\Delta\epsilon^{-4})$. $\square$

### B.3 Proof of Theorem 4.14

*Proof.* Denote $J_t = \overline{\text{vec}}(U_t V_t^\top)^\top H_t \overline{\text{vec}}(U_t V_t^\top)$, $H_t = \nabla^2 f_v(\overline{\text{vec}}(W_t))$, and $f_v(\overline{\text{vec}}(W_t)) = f(W_t)$. Set $\eta_t = \eta$. Taking the Taylor expansion at $W_t$, we have

$$
\begin{aligned}
f(W_{t+1}) - f(W_t) =& \langle \nabla f(W_t), W_{t+1} - W_t \rangle + \frac{1}{2} \overline{\text{vec}}(W_{t+1} - W_t)^\top \nabla^2 f_v(\overline{\text{vec}}(W_t)) \overline{\text{vec}}(W_{t+1} - W_t) \\
& + \frac{1}{6} \sum_{i,j,k=1}^d [\nabla^3 f_v(\theta)]_{ijk} \overline{\text{vec}}(W_{t+1} - W_t)_i \overline{\text{vec}}(W_{t+1} - W_t)_j \overline{\text{vec}}(W_{t+1} - W_t)_k \\
\leq & -\eta \langle \nabla f(W_t), U_t V_t^\top \rangle + \frac{\eta^2 J_t}{2} + \frac{s\eta^3 \|\overline{\text{vec}}(U_t V_t^\top)\|_F^3}{6} \\
\leq & -\eta \langle \nabla f(W_t), U_t V_t^\top \rangle + \frac{\eta^2 J_t}{2} + \frac{s\eta^3 r^{3/2}}{6} \\
= & -\eta \|\nabla f(W_t)\|_* + \frac{\eta^2 J_t}{2} + \frac{s\eta^3 r^{3/2}}{6}
\end{aligned}
\tag{15}
$$

where $\theta \in \mathbb{R}^{mn}$ can be some vectors between $\overline{\text{vec}}(W_t)$ and $\overline{\text{vec}}(W_{t+1})$, and the first inequality is due to Assumption 4.13.

Denote $J = \frac{1}{T} \sum_{t=0}^{T-1} J_t$. Summing over $t = 0, 1, \ldots, T-1$, we can get

$$
\frac{1}{T} \sum_{t=0}^{T-1} \|\nabla f(W_t)\|_* \leq \frac{f(W_0) - f(W_T)}{T\eta} + \frac{\eta J}{2} + \frac{s\eta^2 r^{3/2}}{6}.
$$

$\square$

In experiments, we find that usually $J_t > 0$, thus, if $J \gtrsim \sqrt{s\epsilon}\, r^{3/4}$, and $\eta = \Theta\left(\sqrt{\frac{\Delta}{JT}}\right)$, then

$$
\frac{1}{T} \sum_{t=0}^{T-1} \|\nabla f(W_t)\|_* \leq \left( \sqrt{\frac{J\Delta}{T}} + \frac{sr^{3/2}\Delta}{JT} \right).
$$

Consequently, it is possible that Muon can find an $\epsilon$-nuclear norm stationary point of $f$ with a complexity of $O(J\Delta\epsilon^{-2})$.

Otherwise, if $J \lesssim \sqrt{s\epsilon}\, r^{3/4}$ and $\eta = \Theta\left(\sqrt[3]{\frac{\Delta}{sTr^{3/2}}}\right)$, then

$$
\frac{1}{T} \sum_{t=0}^{T-1} \|\nabla f(W_t)\|_* \leq O\left( \frac{\Delta^{2/3} s^{1/3} r^{1/2}}{T^{2/3}} \right).
$$

Consequently, it is possible that Muon can find an $\epsilon$-nuclear norm stationary point of $f$ with a complexity of $O(s^{1/2} r^{3/4} \Delta \epsilon^{-3/2})$.

### B.4 Proof of Theorem 4.15

*Proof.* Denote $d = mn$, $J_t = \overline{\text{vec}}(U_t V_t^\top)^\top H_t \overline{\text{vec}}(U_t V_t^\top)$, $H_t = \nabla^2 f_v(\overline{\text{vec}}(W_t))$, and $f_v(\overline{\text{vec}}(W_t)) = f(W_t)$. Set $\eta_t = \eta$. Taking the Taylor expansion at $W_t$, we have

$$
\begin{aligned}
& \mathbb{E}[f(W_{t+1}) - f(W_t)] \\
= & \mathbb{E}[\langle \nabla f(W_t), W_{t+1} - W_t \rangle + \frac{1}{2} \overline{\text{vec}}(W_{t+1} - W_t)^\top \nabla^2 f_v(\overline{\text{vec}}(W_t)) \overline{\text{vec}}(W_{t+1} - W_t) \\
& + \frac{1}{6} \sum_{i,j,k=1}^d [\nabla^3 f_v(\theta)]_{ijk} \overline{\text{vec}}(W_{t+1} - W_t)_i \overline{\text{vec}}(W_{t+1} - W_t)_j \overline{\text{vec}}(W_{t+1} - W_t)_k]
\end{aligned}
$$

$$\leq \mathbb{E}\left[-\eta\langle\nabla f(W_t), U_t V_t^\top\rangle + \frac{\eta^2 J_t}{2} + \frac{s\eta^3\|\overline{\text{vec}}(U_t V_t^\top)\|_F^3}{6}\right]$$

$$\leq \mathbb{E}\left[-\eta\langle M_t, U_t V_t^\top\rangle + \eta\langle M_t - \nabla f(W_t), U_t V_t^\top\rangle + \frac{\eta^2 J_t}{2}\right] + \frac{s\eta^3 r^{3/2}}{6}$$

$$\leq \mathbb{E}\left[-\eta\|M_t\|_* + \eta\|M_t - \nabla f(W_t)\|_*\|U_t V_t^\top\|_{\text{op}} + \frac{\eta^2 J_t}{2}\right] + \frac{s\eta^3 r^{3/2}}{6}$$

$$\leq \mathbb{E}\left[-\eta\|\nabla f(W_t)\|_* + 2\eta\|M_t - \nabla f(W_t)\|_* + \frac{\eta^2 J_t}{2}\right] + \frac{s\eta^3 r^{3/2}}{6}$$

where $\theta \in \mathbb{R}^d$ can be some vectors between $\overline{\text{vec}}(W_t)$ and $\overline{\text{vec}}(W_{t+1})$, the first inequality is due to Assumption 4.13.

Then, we need to analyze and bound the error $\|\nabla f(W_t) - M_t\|_*$. Note that $M_0 = G_0$, $M_t = \beta M_{t-1} + (1-\beta)G_t = (1-\beta)\sum_{i=1}^t \beta^{t-i} G_i + \beta^t G_0$. We can define $C_0 = \nabla f(W_0)$, and when $t > 0$, $C_t = \beta C_{t-1} + (1-\beta)\nabla f(W_t) = (1-\beta)\sum_{i=1}^t \beta^{t-i}\nabla f(W_i) + \beta^t \nabla f(W_0)$. Then, according to Lemma A.4, we have

$$\mathbb{E}[\|C_t - M_t\|_*] \leq \sqrt{r}\mathbb{E}[\|C_t - M_t\|_F] \leq \sqrt{\frac{1-\beta}{1+\beta}}\frac{\sigma\sqrt{r}}{\sqrt{B}} + \beta^t\frac{\sigma\sqrt{r}}{\sqrt{B}}$$

Suppose the SVD of $\nabla f(W_t) - \nabla f(W_{t+1})$ is $\nabla f(W_t) - \nabla f(W_{t+1}) = U_{g,t}S_{g,t}V_{g,t}^\top$. Then, we have

$$\|\nabla f(W_t) - \nabla f(W_{t+1})\|_*$$
$$=\|U_{g,t}^\top[\nabla f(W_t) - \nabla f(W_{t+1})]V_{g,t}\|_*$$
$$=\text{tr}\left(U_{g,t}^\top[\nabla f(W_t) - \nabla f(W_{t+1})]V_{g,t}\right)$$
$$=\text{tr}\left((U_{g,t}V_{g,t}^\top)^\top[\nabla f(W_t) - \nabla f(W_{t+1})]\right)$$

For any $i \in [d]$, taking the Taylor expansion of $\overline{\text{vec}}(\nabla f(W_{t+1}))_i$ at $W_t$, we have

$$[\overline{\text{vec}}(\nabla f(W_t) - \nabla f(W_{t+1}))]_i$$
$$=\sum_{j=1}^d [\nabla^2 f_v(\overline{\text{vec}}(W_t))]_{ij}\overline{\text{vec}}(W_t - W_{t+1})_j + \frac{1}{2}\sum_{j,k=1}^d [\nabla^3 f_v(\theta_i)]_{ijk}\overline{\text{vec}}(W_t - W_{t+1})_j\overline{\text{vec}}(W_t - W_{t+1})_k$$
$$=\eta\sum_{j=1}^d [\nabla^2 f_v(\overline{\text{vec}}(W_t))]_{ij}\overline{\text{vec}}(U_t V_t^\top)_j + \frac{\eta^2}{2}\sum_{j,k=1}^d [\nabla^3 f_v(\theta_i)]_{ijk}\overline{\text{vec}}(U_t V_t^\top)_j\overline{\text{vec}}(U_t V_t^\top)_k$$

where $\theta_i \in \mathbb{R}^d$ can be some vectors between $\overline{\text{vec}}(W_t)$ and $\overline{\text{vec}}(W_{t+1})$.

Denote $\hat{J}_t = \overline{\text{vec}}(U_{g,t}V_{g,t}^\top)^\top\nabla^2 f_v(\overline{\text{vec}}(W_t))\overline{\text{vec}}(U_t V_t^\top)$ , we have

$$\|\nabla f(W_{t+1}) - \nabla f(W_t)\|_*$$
$$=\text{tr}\left((U_{g,t}V_{g,t}^\top)^\top[\nabla f(W_t) - \nabla f(W_{t+1})]\right)$$
$$=\eta\sum_{i,j=1}^d \overline{\text{vec}}(U_{g,t}V_{g,t}^\top)_i\nabla^2 f_v(\overline{\text{vec}}(W_t))]_{ij}\overline{\text{vec}}(U_t V_t^\top)_j$$
$$+\frac{\eta^2}{2}\sum_{i,j,k=1}^d [\nabla^3 f_v(\theta_i)]_{ijk}\overline{\text{vec}}(U_{g,t}V_{g,t}^\top)_i\overline{\text{vec}}(U_t V_t^\top)_j\overline{\text{vec}}(U_t V_t^\top)_k$$
$$\leq \eta\hat{J}_t + \frac{s\eta^2 r^{3/2}}{2}$$

where the last inequality is due to Assumption 4.13.

Then, for $t > 0$, we have

$$
\begin{aligned}
&\mathbb{E}[\|\nabla f(W_t) - C_t\|_*] \\
=&\mathbb{E}[\|\nabla f(W_t) - (\beta C_{t-1} + (1-\beta)\nabla f(W_t))\|_*] \\
=&\mathbb{E}[\beta\|\nabla f(W_t) - C_{t-1}\|_*] \\
\leq&\mathbb{E}[\beta\|\nabla f(W_{t-1}) - C_{t-1}\|_* + \beta\|\nabla f(W_{t-1}) - \nabla f(W_t)\|_*] \\
\leq&\mathbb{E}[\beta\|\nabla f(W_{t-1}) - C_{t-1}\|_* + \beta\eta\hat{J}_{t-1} + \frac{s\eta^2 r^{3/2}\beta}{2}] \\
\leq&\sum_{i=0}^{t-1}\beta^{t-i}\eta\mathbb{E}[\hat{J}_i] + \frac{s\eta^2 r^{3/2}\beta}{2(1-\beta)}
\end{aligned}
$$

Thus, we have

$$
\begin{aligned}
\mathbb{E}[\|\nabla f(W_t) - M_t\|_*] \leq&\mathbb{E}[\|C_t - M_t\|_* + \|\nabla f(W_t) - C_t\|_*] \\
\leq&\sqrt{\frac{1-\beta}{1+\beta}}\frac{\sigma\sqrt{r}}{\sqrt{B}} + \beta^t\frac{\sigma\sqrt{r}}{\sqrt{B}} + \sum_{i=0}^{t-1}\beta^{t-i}\eta\mathbb{E}[\hat{J}_i] + \frac{s\eta^2 r^{3/2}\beta}{2(1-\beta)},
\end{aligned}
$$

and

$$
\begin{aligned}
&\mathbb{E}[f(W_t) - f(W_{t+1})] \\
\geq&\mathbb{E}[\eta\|\nabla f(W_t)\|_* - \frac{J_t}{2}\eta^2 - 2\eta\|\nabla f(W_t) - M_t\|_* - \frac{s\eta^3 r^{3/2}}{6} \\
\geq&\mathbb{E}[\eta\|\nabla f(W_t)\|_* - \frac{J_t}{2}\eta^2 - \sum_{i=0}^{t-1}2\beta^{t-i}\eta^2\hat{J}_i] - \frac{s\eta^3 r^{3/2}}{6} - 2\eta\sqrt{\frac{1-\beta}{1+\beta}}\frac{\sigma\sqrt{r}}{\sqrt{B}} - 2\eta\beta^t\frac{\sigma\sqrt{r}}{\sqrt{B}} - \frac{s\eta^3 r^{3/2}\beta}{1-\beta}.
\end{aligned}
$$

Denote $\hat{J} = \frac{1}{T}\sum_{t=0}^{T-2}\hat{J}_t$, $J = \frac{1}{T}\sum_{t=0}^{T-1}J_t$ Summing over $t = 0, 1, \ldots, T-1$, we can get

$$
\begin{aligned}
\frac{1}{T}\sum_{t=0}^{T-1}\mathbb{E}[\|\nabla f(W_t)\|_*] \leq&\frac{1}{T\eta}\mathbb{E}[f(W_0) - f(W_T)] + \frac{\eta\sum_{t=0}^{T-1}\mathbb{E}[J_t]}{2T} + \frac{2\sigma\sqrt{r(1-\beta)}}{\sqrt{(1+\beta)B}} + \frac{2\sigma\sqrt{r}}{(1-\beta)T\sqrt{B}} \\
&+ 2\frac{1}{T}\sum_{t=0}^{T-1}\sum_{i=0}^{t-1}2\beta^{t-i}\eta\mathbb{E}[\hat{J}_i] + O(sr^{3/2}\eta^2/(1-\beta)) \\
\leq&\frac{1}{T\eta}\mathbb{E}[f(W_0) - f(W_T)] + \frac{\eta\mathbb{E}[J]}{2} + \frac{2\sigma\sqrt{r(1-\beta)}}{\sqrt{(1+\beta)B}} + \frac{2\sigma\sqrt{r}}{(1-\beta)T\sqrt{B}} \\
&+ \frac{2\eta\beta\mathbb{E}[\hat{J}]}{1-\beta} + O(sr^{3/2}\eta^2/(1-\beta))
\end{aligned}
$$

Denote $J^* = \max\{\mathbb{E}[\hat{J}], \mathbb{E}[J]\}$, we can get

$$
\frac{1}{T}\sum_{t=0}^{T-1}\mathbb{E}[\|\nabla f(W_t)\|_*] \leq \frac{\Delta}{T\eta} + \frac{J^*\eta}{2} + \frac{2\sigma\sqrt{r(1-\beta)}}{\sqrt{(1+\beta)B}} + \frac{2\sigma\sqrt{r}}{(1-\beta)T\sqrt{B}} + \frac{2\eta\beta J^*}{1-\beta} + O\left(\frac{sr^{3/2}\eta^2}{1-\beta}\right).
$$

$\square$

In experiments, we find that usually $J_t > 0$, thus, if $J^* \gtrsim r^{1/2}\sigma\Delta^{-1}\epsilon$, $B = 1$, $\eta = \sqrt{\frac{(1-\beta)\Delta}{TJ^*}}$, $1-\beta = \min\{\frac{\sqrt{J^*\Delta}}{\sigma\sqrt{rT}}, 1\}$, then

$$
\frac{1}{T}\sum_{t=0}^{T-1}\mathbb{E}[\|\nabla f(W_t)\|_*] \leq O\left(\sqrt[4]{\frac{rJ^*\Delta\sigma^2}{T}} + \sqrt{\frac{J^*\Delta}{T}} + \frac{r\sigma^2}{\sqrt{J^*\Delta T}} + \frac{sr^{3/2}\Delta}{TJ^*}\right).
$$

Consequently, it is possible Muon can find an $\epsilon$-nuclear norm stationary point of $f$ with a complexity of $O(rJ^*\sigma^2\Delta\epsilon^{-4})$.

Otherwise, if $J \lesssim r^{1/2}\sigma\Delta^{-1}\epsilon$, $B = 1$, $1 - \beta = \Theta(r^{-1}\sigma^{-2}\epsilon^2)$, $\eta = \Theta\left(r^{3/2}\sigma^{-3}\Delta\epsilon^2\right)$ $T = \Theta(\sigma^3 r^{3/2}\epsilon^{-3})$ , then

$$\frac{1}{T}\sum_{t=0}^{T-1}\|\nabla f(W_t)\|_* \leq O(\epsilon).$$

Thus, it is possible that Muon can find an $\epsilon$-nuclear norm stationary point of $f$ with a complexity of $O(\sigma^3 r^{3/2}\epsilon^{-3})$.

## C  Star-Convex

### C.1  Proof of Theorem 4.5

*Proof.* **Constant stepsize**

First, by the $L$ Frobenius norm Lipschitz smoothness of $f$, we have

$$f(W_{t+1}) \leq f(W_t) + \langle \nabla f(W_t), W_{t+1} - W_t \rangle + \frac{L}{2}\|W_{t+1} - W_t\|_F^2$$

$$\leq f(W_t) - \eta_t\|\nabla f(W_t)\|_* + \frac{rL\eta_t^2}{2} \tag{16}$$

Then, according to (20), we have

$$-\|\nabla f(W_t)\|_* \leq -\frac{f(W_t) - f^*}{D_{\text{op}}}$$

Set $\eta_t = \eta$ and combine (16) and (20). We have

$$f(W_{t+1}) - f^* \leq \left(1 - \frac{\eta}{D_{\text{op}}}\right)(f(W_t) - f^*) + \frac{rL\eta^2}{2},$$

and

$$f(W_T) - f^* \leq \left(1 - \frac{\eta}{D_{\text{op}}}\right)^T (f(W_0) - f^*) + \sum_{t=0}^{T-1}\left(1 - \frac{\eta}{D_{\text{op}}}\right)^{T-1-t}\frac{rL\eta^2}{2}$$

$$\leq \left(1 - \frac{\eta}{D_{\text{op}}}\right)^T (f(W_0) - f^*) + \frac{rLD_{\text{op}}\eta}{2}.$$

Thus, to reach the precision $f(W_T) - f^* \leq \epsilon$, we can take $\eta = O(\frac{\epsilon}{rLD_{\text{op}}})$, and the complexity is $O(rLD_{\text{op}}^2\epsilon^{-1}\log\frac{\Delta}{\epsilon})$, where $\Delta = f(W_0) - f^*$. $\qquad\square$

### C.2  Proof of Theorem 4.6

*Proof.* **Adaptive stepsize**

By the $L$ spectral norm Lipschitz smoothness of $f$ and setting $\eta_t = \frac{\|\nabla f(W_t)\|_*}{rL}$, we have

$$f(W_{t+1}) \leq f(W_t) + \langle \nabla f(W_t), W_{t+1} - W_t \rangle + \frac{L}{2}\|W_{t+1} - W_t\|_F^2$$

$$\leq f(W_t) - \eta_t\|\nabla f(W_t)\|_* + \frac{rL\eta_t^2}{2}$$

$$= f(W_t) - \frac{\|\nabla f(W_t)\|_*^2}{2rL}. \tag{17}$$

From the star-convex property, we have

$$\Delta_t := f(W_t) - f^* \le \langle \nabla f(W_t), W_t - W^* \rangle \le \|\nabla f(W_t)\|_* \|W_t - W^*\|_{\mathrm{op}}. \tag{18}$$

Combining Equation (17) and Equation (18), we obtain

$$\Delta_{t+1} \le \Delta_t - \frac{1}{2rL\|W_t - W^*\|_{\mathrm{op}}^2}\Delta_t^2.$$

Then we have

$$\frac{1}{\Delta_{t+1}} \ge \frac{1}{\Delta_t} + \frac{\Delta_t}{2rL\|W_t - W^*\|_{\mathrm{op}}^2 \Delta_{t+1}}.$$

Using $\|W_t - W^*\|_{\mathrm{op}} \le D_{\mathrm{op}}$, we can sum above inequality to obtain

$$\frac{1}{\Delta_t} \ge \frac{1}{\Delta_0} + \sum_{i=1}^{t} \frac{\Delta_{i-1}}{2rLD_{\mathrm{op}}^2 \Delta_i} \ge \frac{1}{\Delta_0} + \frac{t}{2rLD_{\mathrm{op}}^2},$$

where we use $\frac{\Delta_{i-1}}{\Delta_i} \ge 1$.

After rearranging terms, we obtain

$$f(W_t) - f^* \le \frac{2rL(f(W_0) - f^*)D_{\mathrm{op}}^2}{2rLD_{\mathrm{op}}^2 + t(f(W_0) - f^*)}.$$

$\square$

### C.3  Proof of Theorem 4.9

*Proof.* **Constant stepsize**

First, by the $L_*$ spectral norm Lipschitz smoothness of $f$, we have

$$f(W_{t+1}) \le f(W_t) + \langle \nabla f(W_t), W_{t+1} - W_t \rangle + \frac{L_*}{2}\|W_{t+1} - W_t\|_{\mathrm{op}}^2$$

$$= f(W_t) - \eta_t \|\nabla f(W_t)\|_* + \frac{L_*\eta_t^2}{2} \tag{19}$$

Then, from the star-convex condition, we obtain

$$f(W_t) \le f^* + \langle \nabla f(W_t), W_t - W^* \rangle$$
$$\le f^* + \|\nabla f(W_t)\|_* \|W_t - W^*\|_{\mathrm{op}}$$
$$\le f^* + \|\nabla f(W_t)\|_* D_{\mathrm{op}}$$
$$-\|\nabla f(W_t)\|_* \le -\frac{f(W_t) - f^*}{D_{\mathrm{op}}} \tag{20}$$

where we apply the Cauchy–Schwarz inequality in the second inequality, and in the third inequality we use Assumption 4.4.

Set $\eta_t = \eta$ and combine (19) and (20). We have

$$f(W_{t+1}) - f^* \le \left(1 - \frac{\eta}{D_{\mathrm{op}}}\right)(f(W_t) - f^*) + \frac{\eta^2 L_*}{2},$$

and

$$f(W_T) - f^* \le \left(1 - \frac{\eta}{D_{\mathrm{op}}}\right)^T (f(W_0) - f^*) + \sum_{t=0}^{T-1} \left(1 - \frac{\eta}{D_{\mathrm{op}}}\right)^{T-1-t} \frac{\eta^2 L_*}{2}$$

$$\leq \left(1 - \frac{\eta}{D_{\mathrm{op}}}\right)^T (f(W_0) - f^*) + \frac{\eta L_* D_{\mathrm{op}}}{2}.$$

Thus, to reach the precision $f(W_T) - f^* \leq \epsilon$, we can take $\eta = O(\frac{\epsilon}{L_* D_{\mathrm{op}}})$, and the complexity is $O(L_* D_{\mathrm{op}}^2 \epsilon^{-1} \log \frac{\Delta}{\epsilon})$, where $\Delta = f(W_0) - f^*$. $\qquad\square$

### C.4  Proof of Theorem 4.10

*Proof.* **Adaptive stepsize**

By the $L_*$ spectral norm Lipschitz smoothness of $f$ and setting $\eta_t = \frac{\|\nabla f(W_t)\|_*}{L_*}$, we have

$$
\begin{aligned}
f(W_{t+1}) &\leq f(W_t) + \langle \nabla f(W_t), W_{t+1} - W_t \rangle + \frac{L_*}{2} \|W_{t+1} - W_t\|_{\mathrm{op}}^2 \\
&= f(W_t) - \eta_t \|\nabla f(W_t)\|_* + \frac{L_* \eta_t^2}{2} \\
&= f(W_t) - \frac{\|\nabla f(W_t)\|_*^2}{2L_*}.
\end{aligned}
\tag{21}
$$

From the star-convex property, we have

$$\Delta_t := f(W_t) - f^* \leq \langle \nabla f(W_t), W_t - W^* \rangle \leq \|\nabla f(W_t)\|_* \|W_t - W^*\|_{\mathrm{op}}. \tag{22}$$

Combining Equation (21) and Equation (22), we obtain

$$\Delta_{t+1} \leq \Delta_t - \frac{1}{2L_* \|W_t - W^*\|_{\mathrm{op}}^2} \Delta_t^2.$$

Then we have

$$\frac{1}{\Delta_{t+1}} \geq \frac{1}{\Delta_t} + \frac{\Delta_t}{2L_* \|W_t - W^*\|_{\mathrm{op}}^2 \Delta_{t+1}}.$$

Using $\|W_t - W^*\|_{\mathrm{op}} \leq D_{\mathrm{op}}$, then we can sum above inequality to obtain

$$\frac{1}{\Delta_t} \geq \frac{1}{\Delta_0} + \sum_{i=1}^t \frac{\Delta_{i-1}}{2L_* D_{\mathrm{op}}^2 \Delta_i} \geq \frac{1}{\Delta_0} + \frac{t}{2L_* D_{\mathrm{op}}^2},$$

where we use $\frac{\Delta_{i-1}}{\Delta_i} \geq 1$.

After rearranging terms, we obtain

$$f(W_t) - f^* \leq \frac{2L_*(f(W_0) - f^*) D_{\mathrm{op}}^2}{2L_* D_{\mathrm{op}}^2 + t(f(W_0) - f^*)}.$$

$\qquad\square$

### C.5  Proof of Theorem 4.16

*Proof.* According to Equation (15), we have

$$f(W_{t+1}) - f(W_t) \leq -\eta \|\nabla f(W_t)\|_* + \frac{\eta^2 J_t}{2} + \frac{s\eta^3 r^{3/2}}{6} \tag{23}$$

Moreover, according to (20), we have

$$-\|\nabla f(W_t)\|_* \leq -\frac{f(W_t) - f^*}{D_{\mathrm{op}}}$$

Combine (23) and (20), we can obtain

$$f(W_{t+1}) - f^* \le \left(1 - \frac{\eta}{D_{\text{op}}}\right)(f(W_t) - f^*) + \frac{\eta^2 J_t}{2} + \frac{s\eta^3 r^{3/2}}{6}.$$

Then, we have

$$f(W_T) - f^* \le \left(1 - \frac{\eta}{D_{\text{op}}}\right)^T (f(W_0) - f^*) + \sum_{t=0}^{T-1} \left(1 - \frac{\eta}{D_{\text{op}}}\right)^{T-1-t} \frac{\eta^2 J_t}{2} + \sum_{t=0}^{T-1} \left(1 - \frac{\eta}{D_{\text{op}}}\right)^{T-1-t} \frac{s\eta^3 r^{3/2}}{6}$$

$$\le \left(1 - \frac{\eta}{D_{\text{op}}}\right)^T (f(W_0) - f^*) + \sum_{t=0}^{T-1} \left(1 - \frac{\eta}{D_{\text{op}}}\right)^{T-1-t} \frac{\eta^2 J_t}{2} + \frac{s\eta^2 D_{\text{op}} r^{3/2}}{6}.$$

Denote $\tilde{J} = \frac{1}{T} \sum_{t=0}^{T-1} \left(1 - \frac{\eta}{D_{\text{op}}}\right)^{T-1-t} J_t$. We have

$$f(W_T) - f^* \le \left(1 - \frac{\eta}{D_{\text{op}}}\right)^T \Delta + \frac{\eta^2 \tilde{J} T}{2} + \frac{s\eta^2 D_{\text{op}} r^{3/2}}{6}.$$

$\square$

Thus, the convergence complexity of Muon can be depending on $\tilde{J}$, which can be viewed as a weighted average of $J_t$.

In experiments, we found that usually $J_t > 0$. Note that if $f$ is strict convex and if $U_t V_t^\top \ne 0$, then $J_t$ is strictly larger than 0.

Thus, if $\tilde{J} \gtrsim s^{1/2} D_{\text{op}}^{-1/2} r^{3/4} \epsilon^{1/2}$ and $\eta = \min\left\{\frac{D_{\text{op}}}{T} \log\left(\frac{T\Delta}{D_{\text{op}}^2 \tilde{J}}\right), D_{\text{op}}\right\}$. We have

$$f(W_T) - f^* \le \frac{D_{\text{op}}^2 \tilde{J}}{T} + \frac{D_{\text{op}}^2 \tilde{J}}{2T}\left[\log\left(\frac{T\Delta}{D_{\text{op}}^2 \tilde{J}}\right)\right]^2 + \frac{sr^{3/2} D_{\text{op}}^3}{6T^2}\left[\log\left(\frac{T\Delta}{D_{\text{op}}^2 \tilde{J}}\right)\right]^2$$

$$\le \tilde{O}\left(\frac{D_{\text{op}}^2 \tilde{J}}{T}\right).$$

Consequently, it is possible that Muon can reach the precision $f(W_T) - f^* \le \epsilon$ with a complexity of $\tilde{O}(\tilde{J} D_{\text{op}}^2 \epsilon^{-1})$.

Otherwise, if $\tilde{J} \lesssim s^{1/2} D_{\text{op}}^{-1/2} r^{3/4} \epsilon^{1/2}$ and $\eta = \min\left\{\frac{D_{\text{op}}}{T} \log\left(\frac{T^2 \Delta}{D_{\text{op}}^3 s r^{3/2}}\right), D_{\text{op}}\right\}$. We have

$$f(W_T) - f^* \le \frac{sr^{3/2} D_{\text{op}}^3}{T^2} + \frac{sr^{3/2} D_{\text{op}}^3}{6T^2}\left[\log\left(\frac{T^2 \Delta}{D_{\text{op}}^3 s r^{3/2}}\right)\right]^2$$

$$\le \tilde{O}\left(\frac{sr^{3/2} D_{\text{op}}^3}{T^2}\right).$$

Consequently, it is possible that Muon can reach the precision $f(W_T) - f^* \le \epsilon$ with a complexity of $\tilde{O}(s^{1/2} r^{3/4} D_{\text{op}}^{3/2} \epsilon^{-1/2})$.

## D  Low-rank Structure of Real-World Data Matrices

In addition to demonstrating the low-rank property of MNIST in Figure 2, we also test and verify similar low-rank structures on the CIFAR-10 (Krizhevsky et al., 2009) and Shakespeare datasets[2].

---

[2]https://raw.githubusercontent.com/karpathy/char-rnn/master/data/tinyshakespeare/input.txt

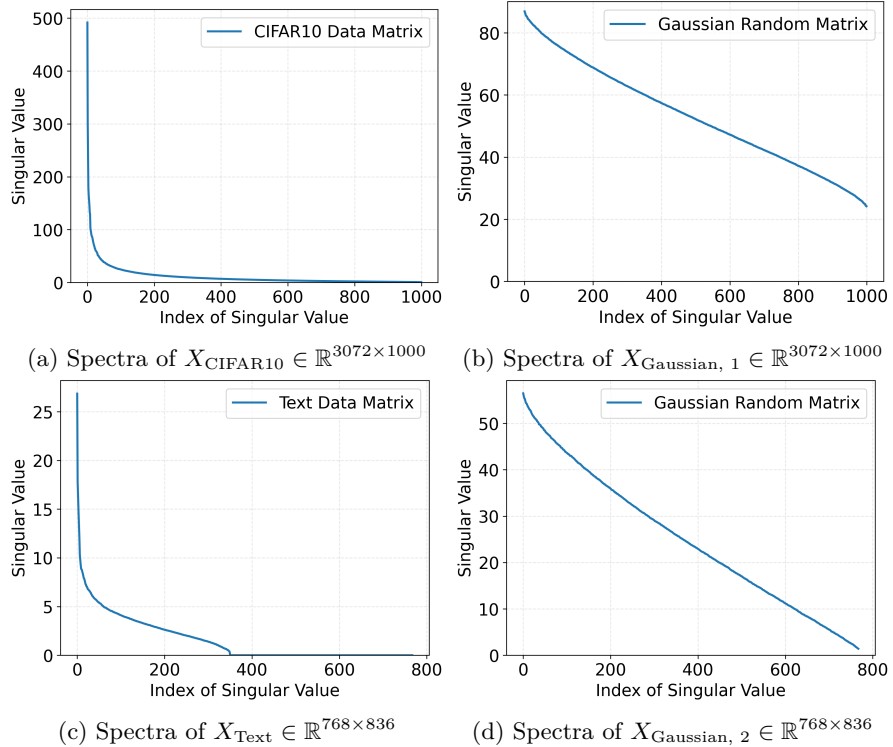

Figure 5: Spectra of $X_{\text{CIFAR10}}$, $X_{\text{Text}}$, $X_{\text{Gaussian, 1}}$ and $X_{\text{Gaussian, 2}}$.

Following a procedure similar to that in Section 4.2.1, for CIFAR-10 we randomly sample 1000 samples from training dataset and construct the matrix $X_{\text{CIFAR10}} \in \mathbb{R}^{3072 \times 1000}$. We then compute its singular values and compare them with those of a Gaussian random matrix of the same size, $X_{\text{Gaussian, 1}} \in \mathbb{R}^{3072 \times 1000}$. The results are shown in Figure 5(a)(b), where we observe that the singular values of $X_{\text{CIFAR10}}$ are significantly more concentrated than $X_{\text{Gaussian, 1}}$.

For the Shakespeare dataset, we take the first 3000 characters and use the RoBERTa (Liu et al., 2019) tokenizer and embedding model to convert the text into an embedding matrix $X_{\text{Text}} \in \mathbb{R}^{768 \times 836}$, where 768 is the embedding dimension and 836 is the token length. We compute its singular values and compare them with those of a Gaussian random matrix of the same size, $X_{\text{Gaussian, 2}} \in \mathbb{R}^{768 \times 836}$. As shown in Figure 5(c)(d), similar to the image datasets (MNIST and CIFAR-10), the text embedding matrix also exhibits a pronounced low-rank structure.

## E  Experimental Settings

Experiments of Figure 2, Figure 3, and Figure 4 are conducted on NVIDIA RTX 6000 Ada GPUs. Experiments of Figure 1 are conducted on Intel(R) Core(TM) i9-14900HX.

In Figure 1, we consider the quadratic function $f(W) = \frac{1}{2}\text{tr}\left((W - W^*)^\top Q(W - W^*)\right)$, $W \in \mathbb{R}^{15 \times 20}$, $Q \in \mathbb{R}^{15 \times 15}$, where $W^*$ is randomly chosen and $Q$ is chosen with an ill condition number. Then, we apply GD and Muon to optimize this function, both start from the $W_0 = 0$ with 4000 iterations. We choose the optimal constant stepsize $\frac{1}{L}$ for GD and choose the stepsize for Muon such that Muon can converge in 4000 iterations with the best function value. For each iteration of both algorithms, we record the difference in function value from the optimum and the spectral norm error to the optimal point. The results are shown in Figure 1(a) and (b). In Figure 1(c), we compare the key constant in convergence analysis under quadratic function with ill-conditioned $Q$. We choose $W^* \sim U(-50, 50)$, which means each element of $W^*$ is i.i.d. random variables drawn from a continuous uniform distribution. Then, we estimate the key constants $D_{\text{op}}^2 L_*$ and $D_{\text{F}}^2 L$, where

we choose the initial point as $W_0 = 0$. We calculate the ratios $\frac{D_{\mathrm{F}}^2 L}{D_{\mathrm{op}}^2 L_*}$ over $10^4$ random samples. The results are shown in Figure 1(c).

In Figure 3, we randomly select a fixed subset of 120 samples from MNIST. We then train a fully connected neural network (Table 1) with three matrix parameters ($W^1 \in \mathbb{R}^{128 \times 784}$, $W^2 \in \mathbb{R}^{64 \times 128}$, $W^3 \in \mathbb{R}^{10 \times 64}$) on this subset using GD and Muon (Algorithm 2). We tuned the learning rates for GD and Muon to their optimal values. During the optimization procedure, we record the loss, $J_t$, $L_t$, $\|\nabla f(W_t)\|_*^2$, and $\|\nabla f(W_t)\|_{\mathrm{F}}^2$. These quantities are computed with respect to $W^2$, i.e., $\nabla f = \nabla_{W^2} f(W_t^1, W_t^2, W_t^3)$ and $\nabla^2 f = \nabla^2_{\overline{\mathrm{vec}}(W^2)\overline{\mathrm{vec}}(W^2)} f_v(\overline{\mathrm{vec}}(W_t^1), \overline{\mathrm{vec}}(W_t^2), \overline{\mathrm{vec}}(W_t^3))$, where $f_v(\overline{\mathrm{vec}}(W_t^1), \overline{\mathrm{vec}}(W_t^2), \overline{\mathrm{vec}}(W_t^3)) = f(W_t^1, W_t^2, W_t^3)$.

Table 1: MLP model used in Figure 3. For every Linear module, the bias term is set as false, so this model contain only matrix parameters.

| Layer Type | Matrix Shape ($m \times n$) |
|---|---|
| Fully Connected + ReLU | $128 \times 784$ |
| Fully Connected + ReLU | $64 \times 128$ |
| Fully Connected | $10 \times 64$ |

In Figure 4, we adopted the code from nanoGPT[3] and trained a 6-layer GPT-2–style model on the Shakespeare character dataset. Model architectural and training hyperparameters can be found in Table 2. We trained the model using AdamW and Muon, respectively. When using Muon, we applied the tricks proposed in Liu et al. (2025a), i.e., using weight decay and adjusting the per-parameter update scale. Muon was used only to optimize the matrix parameters in the non-embedding layers, while the embedding layers and vector parameters were optimized with AdamW.

Table 2: Model architectural and training hyperparameters for the experiments in Figure 4.

| Hyperparameter | Value |
|---|---|
| Number of Layers | 6 |
| Number of Heads | 6 |
| Hidden Dimension | 384 |
| Dropout | 0 |
| Batch Size | 64 |
| Training Steps | 1000 |

---

[3]https://github.com/karpathy/nanoGPT

