# OpenReview forum: "On the Convergence Analysis of Muon"
_TMLR — Decision pending for TMLR_

### Review · Reviewer_wUAo · 2026-04-28

**Summary Of Contributions:**

### Summary

This paper presents a comprehensive theoretical convergence analysis of the Muon optimizer, a recently proposed algorithm specifically designed for matrix-structured parameters in neural networks. The authors analyze Muon's convergence properties under various smoothness assumptions, progressively moving from standard Frobenius norm Lipschitz smoothness to spectral norm Lipschitz smoothness, and further to scenarios without uniform Lipschitz smoothness (relying on bounded $\nabla^3 f$).

The core contribution lies in theoretically characterizing the conditions under which Muon outperforms Gradient Descent (GD). The authors mathematically demonstrate that Muon effectively leverages the low-rank and approximately block-diagonal structures of Hessian matrices. Furthermore, the paper introduces a metric $J$, representing the global averaged Hessian information, to explain Muon's superior convergence rate when the effective rank of the Hessian is low. The theoretical claims are corroborated by empirical results on both quadratic regression tasks and neural network training.

### Strengths
-  Covers a wide range
  of smoothness assumptions.
- Offer genuine
  insight into when and why Muon outperforms GD.
- Validates theoretical predictions on real neural network
  training, measuring J_t, L_t, and gradient norm ratios along the optimization path.

### Weaknesses
- Assumption 4.4, 4.13  are
  restrictive.
- The key quantity J_t depends on the Hessian and
  singular vectors at each step, making it hard to estimate beforehand. The theory’s
  predictive power thus remains partly diagnostic.
- The analysis assumes exact SVD, while practical Muon
  implementations use Newton-Schulz iterations. This gap is acknowledged but not
  theoretically addressed.

**Additional Comments:**

N/A

**Audience:**

Yes

**Audience Explanation:**

The findings of this paper would be of interest to the TMLR audience, which consists of researchers and practitioners in machine learning, particularly those focused on optimization and optimizers.

**Claims And Evidence:**

Yes

**Claims Explanation:**

The theoretical claims are backed by rigorous convergence proofs under multiple
smoothness assumptions, with all technical details provided in the appendix. The
derivations are self-contained and the logical flow from assumptions to rates is
easy to follow.

On the empirical side, the paper does not merely report performance curves; it
actively validates the derived theoretical conditions.

**Requested Changes:**

1. Following weekness 1, the manuscript would be significantly stronger if the authors could provide a theoretical derivation for  Assumption 4.4 (e.g., under specific initialization neighborhoods or stricter smoothness conditions)
2. I wonder is it possible to establish the convergence rate of Algorithm 1 under star convex condition.

---

> ### Author Response · Authors · 2026-06-06
>
> > Assumption 4.4, 4.13 are restrictive. Following weekness 1, the manuscript would be significantly stronger if the authors could provide a theoretical derivation for Assumption 4.4 (e.g., under specific initialization neighborhoods or stricter smoothness conditions)
>
> Thank you for your comment. We justify Assumption 4.4 from three perspectives. First, this assumption is empirically supported by our experimental results (Figure 2(b)). Second, similar assumptions have been widely adopted in literature (e.g., [1, 2]) to establish convergence guarantees under comparable settings as we mentioned in our paper.
>
> Third, to further address this concern, we can remove this assumption under weight decay. We show that the same convergence rate as in Theorem 4.10 can be established without Assumption 4.4 when weight decay is incorporated. We provide the detailed proof for Theorem 4.10, and the proofs of Theorems 4.5, 4.6, and 4.9 under weight decay follow analogously.
> The weight decay update is given by
> $W\_{t+1}=(1-\lambda\_t)W\_t-\eta\_tU\_tV\_t^\top.$
>
> The star convex assumption gives us
> $\langle \nabla f(W\_t), -\lambda\_tW\_t \rangle\leq -\langle \nabla f(W\_t),\lambda\_t W^{\star}\rangle \leq \lambda\_t||\nabla f(W\_t)||\_{\star}||W^{\star}||\_{op}$
> We let $D=4||W^{\star}||\_{op}$ and initialize with $||W\_0||\_{op}\leq D$.
>
> From Lipschitz continuity, we have
> \begin{align} f(W\_{t+1}) &\leq f(W\_t) + \langle \nabla f(W\_t), W\_{t+1}-W\_t\rangle + \frac{L\_{\star}}{2}||W\_{t+1}-W\_t||\_{op}^2 \\\\
> &= f(W\_t) - \eta\_t||\nabla f(W\_t)||\_{\star} + \langle \nabla f(W\_t), -\lambda\_t W\_t\rangle + \frac{L\_{\star}}{2}||\lambda\_t W\_t+\eta\_t U\_tV\_t^\top||\_{op}^2 \\\\
> &\leq f(W\_t) - \eta\_t||\nabla f(W\_t)||\_{\star} + \lambda\_t||\nabla f(W\_t)||\_{\star}||W^{\star}||\_{op} + \frac{L\_{\star}}{2}||\lambda\_t W\_t+\eta\_t U\_tV\_t^\top||\_{op}^2 \\\\
> &\leq f(W\_t) - \frac{||\nabla f(W\_t)||\_{\star}^2}{4L\_{\star}} + \frac{||\nabla f(W\_t)||\_{\star}^2}{4L\_{\star}}\frac{||W^{\star}||\_{op}}{D} + \frac{||\nabla f(W\_t)||\_{\star}^2}{8L\_{\star}} \\\\
> &\leq f(W\_t) - \frac{||\nabla f(W\_t)||\_{\star}^2}{16L\_{\star}}.
> \end{align}
>
> where the third inequality comes from the choices of $\eta\_t, \lambda\_t$, $||W\_t||\_{op}\leq D=4||W^{\star}||\_{op}$(we will prove it later) and Jensen's inequality for last term. The choice of $D$ gives our last inequality. Then following the proof of Theorem 4.10, we can obtain that the convergence rate is $O(L\_{\star}D\_{op}^2\epsilon^{-1})$.
>
> Now we show that the $||W\_t||\_{op}\leq D$ holds for all $t$. We prove it by induction on $t$. The claim holds for $t=0$ by initialization.
> Assume that $||W\_k||\_{op}\le D$ for some $k\ge0$.
> If $\nabla f(W\_k)=0$, then $W\_{k+1}=W\_k$, and the claim follows immediately.
>
> Otherwise,
> \begin{align}
> ||W\_{k+1}||\_{op}
> &=
> ||(1-\lambda\_k)W\_k-\eta\_kU\_kV\_k^\top||\_{op}\\\\
> &\le
> (1-\lambda\_k)||W\_k||\_{op}
> +\eta\_k||U\_kV\_k^\top||\_{op} \\\\
> &\le
> (1-\lambda\_k)D+\eta\_k.
> \end{align}
> Using $\eta\_k=D\lambda\_k$, we obtain
> $||W\_{k+1}||\_{op}\le(1-\lambda\_k)D+D\lambda\_k=D.$
>
> For Assumption 4.13, we justify it from NTK [3] perspective. Neural Tangent Kernel(NTK) shows sufficiently wide neural network behaves locally like a linearized model around its initialization. In this regime, the network output changes approximately linearly with respect to the parameters during training. Consequently, when the loss is a squared loss, the corresponding loss landscape is locally well approximated by a quadratic function. In this case, due to the property of quadratic function, the Assumption 4.13 holds rationally. Hence $||W\_{k+1}||\_{op}\le D$, completing the induction.
>
> References:
>
> [1] Vineet Gupta, Tomer Koren, and Yoram Singer. Shampoo: Preconditioned stochastic tensor optimization. ICML 2018.
>
> [2] Kang An, Yuxing Liu, Rui Pan, Shiqian Ma, Donald Goldfarb, and Tong Zhang. Asgo: Adaptive structured
> gradient optimization. NeurIPS 2025.
>
> [3] Jacot, A., Gabriel, F., \& Hongler, C.. Neural Tangent Kernel: Convergence and Generalization in Neural Networks. NeurIPS 2018.

---

> ### Author Response · Authors · 2026-06-06
>
> >The key quantity $J\_t$ depends on the Hessian and singular vectors at each step, making it hard to estimate beforehand. The theory’s predictive power thus remains partly diagnostic.
>
> Thank you for this meaningful question. We partially agree that $J\_t$ depends on local quantities such as the Hessian and the singular vectors of the update, and therefore may be difficult to estimate accurately before training. Nevertheless, $J\_t$ can still provide useful diagnostic and partially predictive information. In practice, one possible approach is to estimate $J\_t$ using the average value over the first few optimization steps, especially when $J\_t$ does not vary significantly across iterations. This early-stage estimate can then serve as an empirical indicator of Muon.
>
> > The analysis assumes exact SVD, while practical Muon implementations use Newton-Schulz iterations. This gap is acknowledged but not theoretically addressed.
>
> Thank you for your comment. Our goal is to investigate the effect of Muon’s most distinctive component: the orthogonalization of the update. Therefore, we study a simplified version of Muon that uses exact SVD. This design allows us to directly investigate whether the orthogonalized update itself is responsible for the benefits observed in Muon.
>
> In practical implementations, Muon employs Newton–Schulz iterations as an efficient approximation to the exact SVD-based update. Under standard numerical conditions—for example, when the nonzero singular values are sufficiently bounded away from zero—Newton–Schulz iterations closely approximate the exact orthogonalization step [4]. Moreover, alternative numerical methods can also be used to approximate the same SVD-based update.
>
> Since our objective is to understand the role of the orthogonalized update rather than the properties of a particular approximation algorithm, we focus on the exact SVD formulation in our simplified Muon analysis. This allows us to study the core mechanism underlying Muon's behavior without introducing additional approximation-related effects.
>
> >I wonder is it possible to establish the convergence rate of Algorithm 1 under star convex condition.
>
> Thanks for your comment. Yes, we can establish the convergence rate of Algorithm 1 under star convex condition. Here, we provide a proof sketch under Assumption 4.13.
>
> First, according to the Proof of Theorem 4.15, we have
> \begin{align}
>     &E[f(W\_{t+1})-f(W\_{t})]\leq E[-\eta ||\nabla f(W\_t)||\_{\star}+\frac{J\_t}{2}\eta^2+\sum\_{i=0}^{t-1} 2\beta^{t-i} \eta^2 \hat{J}\_{i}]+\frac{s\eta^3r^{3/2}}{6}+2\eta\sqrt{\frac{1-\beta}{1+\beta}}\frac{\sigma\sqrt{r}}{\sqrt{B}}+2\eta \beta^t\frac{\sigma\sqrt{r}}{\sqrt{B}}+\frac{s\eta^3r^{3/2}\beta}{1-\beta}.
> \end{align}
> Moreover, according to (20), we have
> \begin{align}
>     -||\nabla f(W\_t)||\_{\star}&\leq -\frac{f(W\_t)-f^{\star}}{D\_{{op}}}
> \end{align}
>
> Combine these two inequality, we have
> \begin{align}
>     &E[f(W\_{t+1})-f^{\star}]\leq (1-\frac{\eta}{D\_{{op}}})E[f(W\_{t})-f^{\star}]+E[\frac{J\_t}{2}\eta^2+\sum\_{i=0}^{t-1} 2\beta^{t-i} \eta^2 \hat{J}\_{i}]+\frac{s\eta^3r^{3/2}}{6}+2\eta\sqrt{\frac{1-\beta}{1+\beta}}\frac{\sigma\sqrt{r}}{\sqrt{B}}+2\eta \beta^t\frac{\sigma\sqrt{r}}{\sqrt{B}}+\frac{s\eta^3r^{3/2}\beta}{1-\beta}.
> \end{align}
>
> Suppose, there exists a $\bar{J}$ such that for any $t\leq T$, $|J\_t|\leq \bar{J}$ and $|\hat{J}\_t|\leq \bar{J}$, then, we have
> \begin{align}
>         f(W\_T)-f^{\star}&\leq (1-\frac{\eta}{D\_{{op}}})^T\Delta+\frac{\eta^2\bar{J} T}{2}+\frac{\eta^2 \bar{J} T}{1-\beta} +\frac{s\eta^2 D\_{op} r^{3/2}}{6}+2D\_{op} \sqrt{\frac{1-\beta}{1+\beta}}\frac{\sigma\sqrt{r}}{B}+\frac{s D\_{op}\eta^2r^{3/2}}{1-\beta}.
> \end{align}
>
> Thus, for the deterministic setting ($\sigma=0$), the convergence rate is $\tilde{O}(\bar{J}D\_{op}^2\epsilon^{-1})$. And for the stochastic setting, the convergence rate is $\tilde{O}(\bar{J}r\sigma^2D\_{op}^4\epsilon^{-3})$.
>
> [4] Keller Jordan, Yuchen Jin, Vlado Boza, Jiacheng You, Franz Cesista, Laker Newhouse, and Jeremy Bernstein. Muon: An optimizer for hidden layers in neural networks, 2024. URL https://kellerjordan.github.io/posts/muon/

---

### Review · Reviewer_t3SK · 2026-05-01

**Summary Of Contributions:**

This paper analyzes the convergence of the Muon optimizer for matrix-structured parameters, in both a deterministic (Algorithm 2) and a stochastic-with-momentum (Algorithm 1) form, under three smoothness conditions: Frobenius-norm Lipschitz smoothness (Assumption 3.1), spectral-norm Lipschitz smoothness (Assumption 3.2), and a third regime in which only a third-derivative bound is assumed (Assumption 4.13). Under each, complexity bounds are established for reaching an $\epsilon$-nuclear-norm stationary point in both the general nonconvex setting and the star-convex setting, and the resulting rates are compared with those of (S)GD.

The contribution I find most useful is the third regime. The authors introduce a trajectory-averaged quantity $J$, defined as a time average of $\mathrm{vec}(U_t V_t^\top)^\top H_t \, \mathrm{vec}(U_t V_t^\top)$, and obtain a bound that scales with $J$ rather than with a worst-case Lipschitz constant. The paper then argues, both informally and through measured ratios on quadratic regression, MSE linear models on MNIST, CIFAR-10, and Shakespeare embeddings, an MLP on MNIST, and a 6-layer GPT-2 on Shakespeare, that this average is favorable in regimes where the Hessian has approximate low-rank or block-diagonal structure. The paper covers a broader set of smoothness assumptions than is typical for analyses of this algorithm.

**Audience:**

Yes

**Audience Explanation:**

Muon has attracted considerable attention over the past year, both empirically and theoretically. A careful comparison of the conditions under which it provably outperforms GD, particularly via Hessian structure (the constants $L_\star$, $\Vert \Lambda \Vert_\star$, and $J$), should be of interest to the optimization and deep-learning theory readership of TMLR.

**Broader Impact Concerns:**

No.

**Claims And Evidence:**

Yes

**Claims Explanation:**

The convergence rates are stated with clear theorems, the conditions under which Muon outperforms GD are established through the complexity comparisons in Section 4, and these conditions are shown to hold in practice through the experiments on quadratic regression and neural network training.

**Requested Changes:**

Major comments:

1, The corollary in Theorem 4.14 states that Muon attains an $\epsilon$-nuclear-norm stationary point in $O(J\Delta\epsilon^{-2})$ iterations, obtained by setting $\eta = \sqrt{2\Delta/(JT)}$. With this stepsize, the descent inequality has three terms: an initial-gap term, an $\eta J / 2$ term, and a third-order remainder proportional to $s \eta^{2} r^{3/2}$. When I substitute $T = \Theta(J\Delta\epsilon^{-2})$ to make the first two terms of order $\epsilon$, the third-order remainder appears to be of order $s r^{3/2} \epsilon^{2} / J^{2}$. For this to also be at most $\epsilon$, it looks to me as if $J$ would have to be at least on the order of $\sqrt{s\epsilon}\, r^{3/4}$, up to constants; when $J$ is smaller than this, the dominant term seems to be the third-order one, and the actual complexity should scale closer to $s^{1/2} r^{3/4} \Delta \epsilon^{-3/2}$, which is in fact the rate the paper itself derives in the $J \le 0$ branch of the same proof. I may well be missing a step, but if my reading is correct, the simplified statement that Muon achieves $O(J\Delta\epsilon^{-2})$ complexity is only valid in a non-trivial subregime of $J$. Could the authors either qualify the corollary with a lower bound on $J$ of this form, or state the complexity as the maximum of the two regimes? The same comment seems to apply to Theorem B.1 and Theorem 4.15.

2, Section 4.1.1 contains the chain $D_{F} = \Vert W_{0} - W^{\star} \Vert_{F} \le \Vert W_{0} - W^{\star} \Vert_{\star} \le D_{\mathrm{op}}$. The first inequality is fine, but I am not sure about the second: under Assumption 4.4, $D_{\mathrm{op}}$ is an operator-norm radius, and the standard ordering of the three norms is operator $\le$ Frobenius $\le$ nuclear, so the inequality $\Vert W_{0} - W^{\star} \Vert_{\star} \le D_{\mathrm{op}}$ does not in general hold. The bound that does hold, and that seems sufficient for the comparison in this paragraph, is $D_{F} \le \sqrt{r}\, \Vert W_{0} - W^{\star} \Vert_{\mathrm{op}} \le \sqrt{r}\, D_{\mathrm{op}}$, equivalently $D_{F}^{2} \le r D_{\mathrm{op}}^{2}$, which still gives $L D_{F}^{2} \le r L D_{\mathrm{op}}^{2}$ and so the qualitative conclusion that Muon is no better than (S)GD in this regime is preserved. The same point should be checked in Section 4.2.1, where the analogous comparison between $L_{\star} D_{\mathrm{op}}^{2}$ and $L D_{F}^{2}$ is drawn. Could the authors confirm this and adjust the chain in Section 4.1.1 (and other places, if applicable) accordingly?


Minor comments:

1, In Eq. (3) the loss carries a factor of $1/(2B)$, but Eq. (4) writes the Hessian as the Kronecker product of $I_{c}$ with $XX^{\top}$ without any $1/B$. Unless I am miscounting the differentiations, a factor of $1/B$ should be inherited from the loss, and the values $L = \Vert X \Vert_{\mathrm{op}}^{2}$ and $L_{\star} = \Vert X \Vert_{F}^{2}$ stated immediately afterwards would then carry the same $1/B$. This factor is common to both quantities, so the ratios used in the subsequent comparison are unaffected, but the formula itself appears to need the correction for consistency with Eq. (3) and with the more general Hessian computation in Eq. (6).

2, On p. 11 the paper writes that in the single-sample case $J \le L$ "demonstrates the significant advantage of Muon compared to GD." It might be cleaner to state this comparison directly in terms of the two complexity bounds: Muon's bound (in the valid regime of Theorem 4.14) is on the order of $J \Delta \epsilon^{-2}$, while GD's nuclear-stationarity bound, after applying Proposition 3.6, is on the order of $r L \Delta \epsilon^{-2}$, so $J \le L$ already gives a factor-$r$ formal improvement.

3, The stochastic analysis in Appendix B.4 controls the nuclear-norm difference of consecutive gradients by a quantity of the form $\eta \widehat{J}_t$ plus a third-order remainder, where $\widehat{J}_t$ is a cross-Hessian inner product between the singular vectors of the gradient difference and those of the current update. Because this is a cross term rather than a quadratic form, it does not appear to be non-negative in general. The final bound uses $J^{\star} = \max\{\mathbb{E}[\widehat{J}], \mathbb{E}[J]\}$, but I could not see where the case $\widehat{J}_t < 0$ is handled. Could the authors clarify whether the bound implicitly uses $|\widehat{J}_t|$, or whether positivity is being assumed in this regime?

---

> ### Author Response · Authors · 2026-06-06
>
> **Major comments 1**
>
> **Reply**: Thank you for pointing out this issue. We have revised the interpretation of the results in these theorems and now discuss the convergence rates separately for different sub-regimes of $J$ for Theorems 4.14, 4.13 and 4.15. Complete discussions can be found in Appendices B.3, B.4, and C.5.
>
> **Major comments 2**
>
> **Reply**: Thank you for pointing out this typo. Indeed, the inequality we intended to state is $D\_F^2=||W\_0-W^{\star}||\_F^2\leq r||W\_0-W^{\star}||\_{op}^2\leq r D_{op}^2$ and we have corrected this issue in the revised paper.
>
> **Minor comments 1**
>
> **Reply**: Thank you for pointing out this typo. We have corrected this issue in the revised paper.
>
> **Minor comments 2**
>
> **Reply**: Thank you for the suggestion. However, as Reviewer aR1K pointed out in the Additional Comments and as we discussed in Section 4.3.1, the conversion from a Frobenius-norm stationary point to a nuclear-norm stationary point does not always imply a strict factor-(r) improvement. For this reason, we chose not to make such a claim in the paper. Nevertheless, because we at least have $||\cdot||\_F\leq ||\cdot||\_*$, thus, when $J \leq L$, the convergence rate of Muon can be significantly better than GD's.
>
> **Minor comments 3**
>
> **Reply**: Thanks for the comment. We have added the discussion of the convergence behaviors in the two regimes: $J^{\star}>\Theta (r^{1/2}\sigma\Delta^{-1}\epsilon)$  and $J^{\star}< \Theta (r^{1/2}\sigma\Delta^{-1}\epsilon)$.

---

### Review · Reviewer_aR1K · 2026-05-22

**Summary Of Contributions:**

This paper investigates the theoretical convergence properties of the Muon optimizer and its simplified variants under various optimization settings. The primary contribution is establishing non-vacuous convergence guarantees for Muon under both Frobenius norm and spectral norm smoothness assumptions, as well as for a simplified Muon under additional star convexity and bounded parameter assumptions (including a setting without Lipschitzness). Furthermore, through a detailed theoretical comparison with (Stochastic) Gradient Descent, the study successfully demonstrates that Muon outperforms SGD when the Hessian matrix exhibits a low-rank structure.

**Additional Comments:**

There is a critical question regarding the theoretical comparison that needs to be addressed, and I would appreciate it if the authors could revise the paper accordingly to make the narrative more rigorous.

In the comparison with (S)GD, Proposition 3.6 is widely used for the derivation of the convergence rate of (S)GD. However, the matrix norm inequality $ \\|A\\|_* \leq \sqrt{r}\\|A\\|_F$ can be quite loose when $A$ is low rank or when the structure of the singular values of $A$ is complex. This potential looseness might render the theoretical comparison to SGD somewhat unfair or overly pessimistic for SGD. I would appreciate it if the authors could provide a more detailed explanation or justification regarding this approximation, and discuss how it impacts the fairness of the comparison.

**Audience:**

Yes

**Audience Explanation:**

Yes. Muon has recently gained significant attention as a highly effective optimization algorithm for training large-scale neural networks. This work presents a comprehensive set of convergence results for Muon under different theoretical settings and successfully identifies the specific structural conditions (such as low-rank Hessians) where Muon outperforms standard (S)GD. These theoretical insights will be of great interest to researchers in optimization and deep learning within the TMLR community.

**Broader Impact Concerns:**

No.

**Claims And Evidence:**

Yes

**Claims Explanation:**

Yes. The claims are well-supported by the following strengths:

1. The authors provide a solid theoretical convergence analysis for Muon and simplified Muon under Frobenius norm smooth, spectral norm smooth, and even non-smooth cases. They compare it with (S)GD in all cases, showing a comparable convergence rate in the general worst-case scenario and a better performance when the Hessians of $f$ have a blockwise diagonal structure and are relatively "low rank". This theoretical finding aligns well with existing studies and their experimental results, making it highly convincing.

2. I have carefully checked the proofs of the inequalities involving matrix norms and their main convergence results, finding the logical flow to be clear and the derivations to be fundamentally correct.

**Requested Changes:**

1. **Literature Comparison:** Please provide a more comprehensive comparison of the convergence results of Muon with existing studies [1,2,3,4].
2. **Algorithm Differences and Assumptions:** Please explain in more detail the algorithmic differences between the standard Muon and the simplified Muon, and clarify the rationale behind analyzing only the simplified Muon under the star convexity assumption. Additionally, please move the convergence analysis of Muon in the non-smooth case (currently discussed in the Appendix) into the main text and provide further intuitive explanations of these results.
3. **Typo Corrections in Proofs:** Please correct the following typos in the mathematical proofs:
    *   3.1 In the proof of Lemma A.4, in the third inequality right below Eq. (14), it should be $\\sum \\beta\^{2(t-i)}$ instead of $\\sum\beta\^{t-i}$.
    *   3.2 In the proof of Theorem 4.1 (and Theorem 4.7) on Page 20 (and Page 21), after summing over \( t=0,1,\dots,T-1 \), the term involving $\frac{2\beta\sigma\sqrt{r}}{(1-\beta)T\sqrt{B}}$ should be corrected to$\frac{2\sigma\sqrt{r}}{(1-\beta)T\sqrt{B}}$.
    *   3.3 In the proof of Theorem 4.6, it should be $\frac{\Delta_t}{\Delta_{t+1}} \geq 1$ instead of $\frac{\Delta_t}{\Delta_{t-1}} \geq 1$.

[1] Li J, Hong M. A note on the convergence of muon and further[J]. arXiv e-prints, 2025: arXiv: 2502.02900.

[2] Pethick T, Xie W, Antonakopoulos K, et al. Training deep learning models with norm-constrained lmos[J]. arXiv preprint arXiv:2502.07529, 2025.

[3] An K, Liu Y, Pan R, et al. Asgo: Adaptive structured gradient optimization[J]. Advances in Neural Information Processing Systems, 2026, 38: 126775-126814.

[4] Kovalev D. Understanding gradient orthogonalization for deep learning via non-euclidean trust-region optimization[J]. arXiv preprint arXiv:2503.12645, 2025.

---

> ### Author Response · Authors · 2026-06-06
>
> >Please provide a more comprehensive comparison of the convergence results of Muon with existing studies [1,2,3,4].
>
> Thank you for your valuable comment. Below, we provide a more comprehensive comparison of the convergence results for Muon with those reported in prior studies [1–4].
>
> Compared with [1,2,3,4], our work provides a more comprehensive convergence analysis of Muon under various smoothness assumptions (Assumption 3.1, 3.2, 4.11), including scenarios without uniform Lipschitz smoothness (Assumption 4.13).
>
> More specifically, [1] analyzed the Frobenius norm convergence of Muon under Frobenius norm Lipschitz smoothness, while our work establishes the nuclear norm convergence guarantees of Muon under the Frobenius norm Lipschitz smoothness.
>
> In [3], the primary contribution was the proposal and analysis of the ASGO algorithm. Since ASGO shares certain conceptual similarities with Muon, the authors additionally analyzed the convergence of a Simplified Muon (Algorithm 2) under Assumption 4.11 in the nonconvex setting, while in our work, we additionally prove Lemma 4.12 and establish the convergence of Muon under Assumption 4.11 in both nonconvex and star convex settings.
>
> In [2], the authors introduced a general algorithmic framework based on a family of linear minimization oracle (lmo) methods and established convergence guarantees for this family. Muon can be viewed as a special instance of this framework, and its convergence was analyzed under Assumption 3.2. However, Theorems 5.5 and 5.7 in [2] require a vanishing momentum parameter to guarantee convergence, whereas our Corollary 4.8 establish convergence guarantees for Muon with a constant momentum parameter.
>
> Finally, [4] proposed a stochastic non-Euclidean trust-region gradient method with momentum, treating Muon, normalized SGD, and signSGD as special cases. [4] established convergence guarantees for the stochastic non-Euclidean trust-region gradient method in both star-convex and nonconvex settings. Compared with [4], in the star-convex setting, we additionally establish the convergence of Muon with adaptive learning rates, achieving an improved complexity bound with a $\log(\epsilon^{-1})$ factor improvement over the constant-step-size result in [4].
>
> Moreover, compared with [1,2,3,4], we additionally offer detailed comparisons with GD and characterize the conditions under which Muon outperforms GD, and validate these conditions through experiments. Our analysis reveals that Muon can exploit the low-rank structure of Hessian matrices, offering a theoretical explanation for its advantages in structured optimization problems. Therefore, we believe our work presents distinct contributions and insights.
>
> >Please explain in more detail the algorithmic differences between the standard Muon and the simplified Muon, and clarify the rationale behind analyzing only the simplified Muon under the star convexity assumption.
>
> Thank you for this insightful question. The standard Muon and our simplified Muon differ in two main aspects: (i) the use of momentum and (ii) the computation of the SVD update. In standard Muon, the update is obtained by applying Newton–Schulz iterations, a numerical procedure that efficiently approximates the SVD-based step. Since momentum has already been widely adopted in optimization methods, our goal is to investigate the effect of Muon’s most distinctive component: the SVD that orthogonalizes the update. Therefore, we consider a simplified Muon variant without momentum. This design allows us to focus on whether the orthogonalized update itself is responsible for Muon’s benefits. The Newton–Schulz iterations provide an efficient approximation to the exact SVD-based update under standard numerical conditions, e.g., when the nonzero singular values are not too close to zero [1]. Also there are some other numerical method to approximate the exact SVD. Hence to focus on the benefit of SVD, we consider exact SVD in simplified Muon.
>
> We consider the star convexity assumption because our goal is to analyze convergence in function value, rather than merely first-order stationary convergence. This setting allows us to investigate when Muon can leverage the benefit of the SVD orthogonalization and achieve better function value convergence than gradient descent (GD). Therefore, it is sufficient for our purpose to analyze the simplified Muon variant under this assumption, as it focuses on the effect of the SVD orthogonalized update which is the most vital part of Muon.
>
> [1] Keller Jordan, Yuchen Jin, Vlado Boza, Jiacheng You, Franz Cesista, Laker Newhouse, and Jeremy
> Bernstein. Muon: An optimizer for hidden layers in neural networks, 2024. URL https://kellerjordan.github.io/posts/muon/

---

> ### Author Response · Authors · 2026-06-06
>
> >please move the convergence analysis of Muon in the non-smooth case (currently discussed in the Appendix) into the main text and provide further intuitive explanations of these results.
>
> Thanks for your advise, we have moved the convergence analysis of Muon in the non-smooth case into the main text and provide the intuitive explanations of these results in the revised paper.
>
> >Typo Corrections in Proofs
>
> Thank you very much for pointing out these typos. We have corrected them in the revised paper.
>
> >In the comparison with (S)GD, Proposition 3.6 is widely used for the derivation of the convergence rate of (S)GD. However, the matrix norm inequality $||A||\_*\leq\sqrt{r} ||A||\_F$ can be quite loose when \(A\) is low rank or when the structure of the singular values of $A$ is complex. This potential looseness might render the theoretical comparison to SGD somewhat unfair or overly pessimistic for SGD. I would appreciate it if the authors could provide a more detailed explanation or justification regarding this approximation, and discuss how it impacts the fairness of the comparison.
>
>
> Thanks for your comment. In fact, we already acknowledged this issue in the paper. For example, in Section 4.3.1, we explicitly discussed it in the paragraph immediately preceding Equation (7):
>
> “Various studies have shown that the gradients during neural network optimization are also typically low rank. Thus, when we convert the stationary point condition from the Frobenius norm to the nuclear norm, the additional coefficient can be much smaller than $\sqrt{r}$. Therefore, to compare the convergence rates of Muon and GD more precisely and fairly, one should examine the ratios between the nuclear norm and Frobenius norm of their gradients as well as $J_t$ and $L_t$.”
>
> Motivated by this observation, we proposed Equation (7) as a more precise and fair criterion for comparing Muon and GD. Furthermore, our experimental results show that this inequality consistently holds throughout the optimization trajectory, thereby demonstrating the advantage of Muon over GD.

---

### Decision · Action_Editor_wUis · 2026-06-26

**Recommendation:** Accept as is

**Audience:**

Yes

**Audience Explanation:**

Muon is a widely used optimization method for training neural networks, gaining increasing attention due to its efficiency. This paper offers a comprehensive convergence rate analysis for Muon, demonstrating that it can outperform gradient descent under certain low-rank assumptions. These findings are likely to be of significant interest to both the optimization and machine learning communities.

**Claims And Evidence:**

Yes

**Claims Explanation:**

This paper investigates the convergence properties of Muon optimizers under various smoothness assumptions. All reviewers agree that this is a strong submission, featuring interesting results and clear comparisons with existing methods. The analysis effectively highlights the advantages of Muon optimizers over traditional gradient descent, particularly for low-rank models. Notably, the convergence rate is characterized using a trajectory-averaged quantity rather than the worst-case Lipschitz constant, providing deeper insight into the optimizer's behavior. These findings are valuable for understanding the benefits of treating model weights as matrices rather than vectors in neural network training. Additionally, all reviewers comment that the proofs are solid, well-structured, and easy to follow.